# FlexCAD: Unified and Versatile Controllable CAD Generation with Fine-tuned Large Language Models

**Zhanwei Zhang**[1]*, **Shizhao Sun**[2†], **Wenxiao Wang**[3†], **Deng Cai**[1], **Jiang Bian**[2]

[1] State Key Lab of CAD&CG, Zhejiang University

[2] Microsoft Research

[3] School of Software Technology, Zhejiang University

{zhanweizhang, wenxiaowang}@zju.edu.cn

{shizsu, jiabia}@microsoft.com, dengcai@cad.zju.edu.cn

## Abstract

Recently, there is a growing interest in creating computer-aided design (CAD) models based on user intent, known as controllable CAD generation. Existing work offers limited controllability and needs separate models for different types of control, reducing efficiency and practicality. To achieve controllable generation across all CAD construction hierarchies, such as sketch-extrusion, extrusion, sketch, face, loop and curve, we propose FlexCAD, a unified model by fine-tuning large language models (LLMs). First, to enhance comprehension by LLMs, we represent a CAD model as a structured text by abstracting each hierarchy as a sequence of text tokens. Second, to address various controllable generation tasks in a unified model, we introduce a hierarchy-aware masking strategy. Specifically, during training, we mask a hierarchy-aware field in the CAD text with a mask token. This field, composed of a sequence of tokens, can be set flexibly to represent various hierarchies. Subsequently, we ask LLMs to predict this masked field. During inference, the user intent is converted into a CAD text with a mask token replacing the part the user wants to modify, which is then fed into FlexCAD to generate new CAD models. Comprehensive experiments on public dataset demonstrate the effectiveness of FlexCAD in both generation quality and controllability. Code will be available at https://github.com/microsoft/FlexCAD.

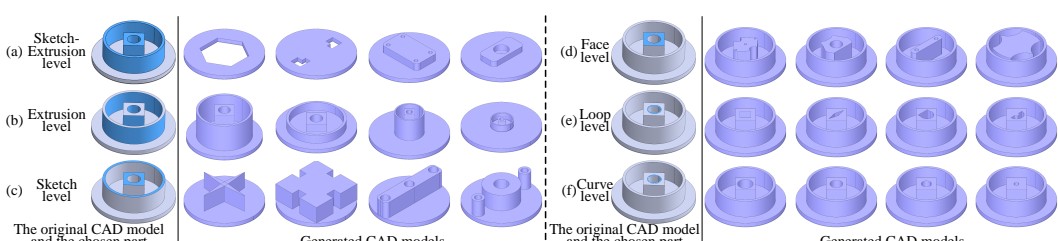

Figure 1: Controllable CAD generation achieved by FlexCAD. In each sub-figure, the left side shows the input: an original CAD model along with the part the user intends to modify (highlighted in blue). The right side displays the output: multiple new CAD models with only the chosen part changed. Users have the flexibility to specify the part in any CAD construction hierarchies, ranging from coarse levels like sketch-extrusion to fine levels like curve (as illustrated from (a) to (f)).

---

* Work done during an internship at Microsoft Research Asia.

† Corresponding author.

# 1 INTRODUCTION

A computer-aided design (CAD) model is a digital representation of a 2D or 3D object. It has been widely used across numerous industries, including architecture, product design and manufacturing, facilitating precise, efficient, and innovative development Ganin et al. (2021); Khan et al. (2024). In commonly used CAD tools like SolidWorks and AutoCAD, *sketch-and-extrude modeling* (SEM) is prevalent. This involves drawing 2D sketches and then extruding them into 3D shapes. Compared to other representations, such as Constructive Solid Geometry (CSG) Yu et al. (2024), B-rep Xu et al. (2024), or voxel Li et al. (2023) and point cloud Khan et al. (2024)-based formats, SEM, incorporating multiple CAD construction hierarchies including *sketch-extrusion*, *extrusion*, *sketch*, *face*, *loop* and *curve* (see Fig. 3(a)), directly illustrates the drawing process of a 3D object. This allows for easy editing and reuse of CAD models, which is essential in CAD development.

Recently, there is an increasing interest in developing generative models to automatically produce SEM of a CAD model[1]. Specifically, DeepCAD Wu et al. (2021) focuses on uncontrollable generation, where a CAD model is generated from a randomly sampled vector. However, providing controllability, *i.e.*, generating CAD models according to user intent, is crucial for the practical application of generative models. To address this, SkexGen Xu et al. (2022) and Hnc-cad Xu et al. (2023) implement disentangled codebooks to offer some levels of control. As each codebook encodes a particular construction hierarchy, their controllability is quite restricted. For instance, SkexGen does not allow selecting a specific sketch for modifications when a CAD model comprises multiple sketches, nor can it handle finer-grained hierarchies such as faces and loops. Hnc-cad lacks control over the topology and geometry of curves. In summary, existing methods face challenges in providing adequate controllability across all CAD construction hierarchies. Additionally, they require separate models to deliver different types of control, which is inefficient and less practical.

The emergence of large language models (LLMs) offers insights for addressing these challenges. First, LLMs have exhibited remarkable success in handling diverse user queries with a single and unified model Chung et al. (2024). This phenomenon not only occurs in natural language tasks but also extends to other areas with domain-specific fine-tuning, such as human motion generation Jiang et al. (2024) and crystal material synthesis Gruver et al. (2024). Second, LLMs might have acquired CAD-related knowledge during the pre-training by learning CAD-specific codes, such as JSCAD codes Makatura et al. (2023). Third, prior to the rise of LLMs, small transformer-based models were explored for tasks like uncontrollable generation and image-to-sketch translation in the 2D sketch domain Ganin et al. (2021), showcasing the possibility of LLMs from a different perspective.

In this work, we introduce FlexCAD, a unified model designed for controllable CAD generation across all hierarchies by fine-tuning LLMs. As shown in Fig. 1, FlexCAD receives the original CAD model along with the part the user wants to modify (highlighted in blue). Here, users can specify the part in any hierarchy. FlexCAD then generates multiple new CAD models, altering only the selected part. To achieve these abilities, first, FlexCAD translates a CAD model into a *concise and structured text* (see Fig. 3). Specifically, in each sketch, the curve type (*e.g.*, a line) is directly represented as textual tokens. The numerical data indicating geometry (*e.g.*, point coordinates in a line) is converted into decimal integers and then into textual tokens. A special token is added to mark the end of each hierarchy. Tokens from the finer-level hierarchy are concatenated to form the representation for the coarser-level hierarchy. We use a similar way to convert each extrusion. Consequently, unlike the one-hot representation used in Xu et al. (2022), FlexCAD provides a concise text representation of a CAD model, facilitating easier processing and understanding by LLMs. Second, FlexCAD introduces a *hierarchy-aware masking* strategy to enable fine-tuning LLMs for various controllable CAD generation tasks (see Fig. 2). During training, we replace a hierarchy-aware field, which contains a sequence of tokens in the CAD text, with a mask token. This field can be set adaptably to reflect various hierarchies. Then, we ask LLMs to predict the masked field. To achieve this, we design prompt templates for all hierarchies, where the mask tokens are tailored to match the corresponding hierarchies. These templates are uniformly sampled at each epoch during the fine-tuning of LLMs. In this way, we ensure that the generation tasks for all hierarchies are learned in a single and unified model. Besides, unlike Xu et al. (2022; 2023) that requires multi-stage training, FlexCAD achieves end-to-end training. During inference, a CAD model is represented as a CAD text with a mask token replacing the part the user wants to change. The masked CAD text is fed into the

---

[1] In the following, we will use CAD model to refer to SEM of a CAD model for brevity.

fine-tuned LLMs to get predictions. After infilling the masked text with these predictions, FlexCAD produces CAD texts that can be rendered into new CAD models. Overall, our contributions are:

- We propose FlexCAD, a unified and versatile model for controllable CAD generation across all hierarchies, including sketch-extrusion, extrusion, sketch, face, loop and curve.
- To the best of our knowledge, FlexCAD is the first to leverage LLMs for controllable CAD generation. It converts a CAD model into a brief, structured text and employs hierarchy-aware masking to fine-tune LLMs for various controllable CAD generation tasks.
- We conduct extensive experiments on public datasets. Despite its simplicity, FlexCAD greatly improves generation quality and controllability, showing its effectiveness on the tasks presented in this work and indicating potential for other CAD generation scenarios.

## 2 RELATED WORK

**CAD Model Generation.** A CAD model represents a 2D or 3D object digitally, with far-reaching applications spanning multiple industries Li et al. (2024b). Existing CAD generation methods can be classified into three categories based on representations of CAD models Khan et al. (2024): constructive solid geometry (CSG), boundary representation (B-rep) and sketch-and-extrude modeling (SEM). CSG combines primitives (*e.g.*, cubes, cylinders, or spheres) via Boolean operations (*e.g.*, union, subtraction or difference) to construct a CSG tree Laidlaw et al. (1986); Du et al. (2018); Kania et al. (2020); Ren et al. (2021); Yu et al. (2022; 2024). B-rep characterizes a CAD model as a graph Ansaldi et al. (1985), consisting of sets of interconnected faces, edges, and vertices Jayaraman et al. (2023); Wang et al. (2022); Jayaraman et al. (2023); Xu et al. (2024).

Our work focuses on SEM, which directly models the drawing process of a CAD model, *i.e.*, drawing 2D curves to make sketches and then extruding them into 3D shapes. There are multiple construction hierarchies in SEM, such as sketch-extrusion, extrusion, sketch, face, loop and curve. Compared to CSG and B-rep, it enables easy editing and reuse of CAD models, which is critical in the CAD design process. For SEM of a CAD model, DeepCAD Wu et al. (2021) utilizes a transformer-based Vaswani et al. (2017) autoencoder for unconditional CAD generation. It merely designates controllable generation as a future application. Nevertheless, offering controllability, *i.e.*, the ability to generate CAD models based on user intents, is critical for enhancing the efficiency of CAD design workflow in practical applications. SkexGen Xu et al. (2022) proposes to disentangle the topology and geometry of sketches and extrusions to provide some levels of control. However, it does not allow choosing a specific sketch, face, loop or extrusion for editing when multiple options exist. Hnc-cad Xu et al. (2023) exploits three codebooks based on VQ-VAE framework Van Den Oord et al. (2017) to control loop, sketch and extrusion level generation, respectively. While it offers finer control, it struggles to manage the topology and geometry in the curve level. To sum up, despite significant progress, existing work face challenges in offering controllability across all construction hierarchies. Besides, they develop separate models to support different types of control.

**Large Language Models (LLMs).** LLMs have shown significant success recently Touvron et al. (2023); Liu et al. (2024). Fine-tuning LLMs has demonstrated notable versatility and efficacy Luo et al. (2024); Chung et al. (2024); Roziere et al. (2023); Ma et al. (2024); Zhang et al. (2024); Li et al. (2024a). Specifically, the fine-tuned LLMs have displayed exceptional success by simultaneously handling various generation tasks within a unified framework Wu et al. (2024); Jiang et al. (2024); Gruver et al. (2024); Zou et al. (2024). Moreover, LLMs are generally pre-trained using extensive high-quality datasets with cross-disciplinary and abundant knowledge Wenzek et al. (2020); Touvron et al. (2023). For example, LLMs may have acquired CAD-related knowledge by learning CAD codes, such as JSCAD codes Makatura et al. (2023). Encouraged by these achievements, we explore the potential of using LLMs to tackle the aforementioned challenges in CAD generation.

## 3 METHODOLOGY

In this section, we introduce FlexCAD, a unified model for controllable CAD generation across all construction hierarchies. As shown in Fig. 1, it receives an original CAD model along with the part the user wants to modify (highlighted in blue), and generates multiple new CAD models with only the selected part altered. To achieve this, as illustrated in Fig. 2, FlexCAD first translates a CAD

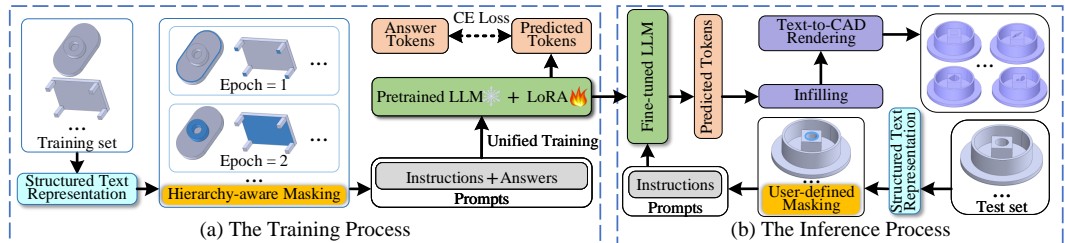

Figure 2: The overall framework of FlexCAD. (a) Training process. Initially, a CAD model is converted into a structured text. Next, a hierarchy-aware masking strategy is proposed to mask a specific field in the text with a special mask token. This field is set differently at each epoch to reflect various hierarchies. Then, LLMs are fine-tuned to predict the masked field. (b) Inference process. The original CAD model is transformed into a structured text with a mask token replacing the part the user wants to change. The fine-tuned LLMs are provided with this masked text to generate diverse predictions, which are then converted into new CAD models by infilling and rendering.

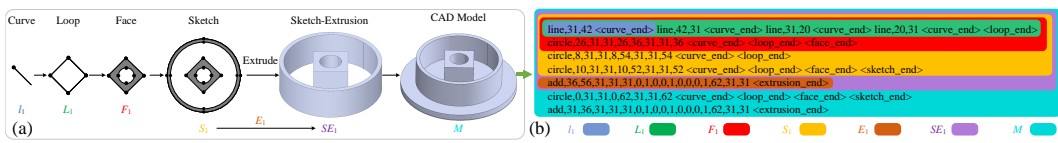

Figure 3: (a) An illustration for construction hierarchies of a CAD model. (b) Structured text representation for the CAD model shown in (a). The colors beneath the texts in (b) are used to indicate the relationship to construction hierarchies depicted in (a), *e.g.*, blue for a curve and green for a loop.

model into a structured text (Sec. 3.1), and then introduces a hierarchy-aware masking strategy to enable fine-tuning LLMs for multiple controllable CAD generation tasks (Sec. 3.2)

## 3.1 REPRESENTING CAD AS STRUCTURED TEXT

Following the conventional definition of SEM of a CAD model Wu et al. (2021); Xu et al. (2022), there are multiple construction hierarchies, as illustrated in Fig. 3(a). A '*curve*', *i.e.*, line $l$, arc $a$, or circle $c$, forms the base level, represented by one, two, or four points respectively. Each point is denoted by its $x$ and $y$ coordinates. A '*loop*' $L$ denotes a closed path, comprising either a single curve (*i.e.*, circle) or multiple curves (*e.g.*, line-arc-line). A '*face*' $F$ is a 2D area, characterized by a single loop or an outer loop with one or multiple inner loops acting as holes. A '*sketch*' $S$ is composed of one or multiple faces, sharing a common extrusion command. An '*extrusion*' $E$ is a command that extends a sketch from a 2D plane into a 3D body. A '*sketch-extrusion*' $SE$ represents a single sketch-extrusion 3D body. A '*CAD model*' $M$ comprises one or multiple $SE$ entities. As shown in Fig. 3(b), we represent a CAD model as a succinct and structured text. Specifically, in each sketch, we start by representing a curve since it is the base level. The curve type (*i.e.*, line, arc or circle) is represented directly as textual tokens. The point coordinates of the curve, which are numerical, are expressed as decimal integers and then converted into textual tokens. This contrasts with Xu et al. (2022) that uses binary representation for point coordinates. For example, when discretizing coordinates into a 64×64 grid, Xu et al. (2022) denotes the center coordinate as $([0, 1, 1, 1, 1, 1], [0, 1, 1, 1, 1, 1])$, while we represent it as $(31, 31)$. Next, the curve is denoted as a sequence of textual tokens, with the first one indicating its type and the others representing point coordinates (the text with a blue background in Fig. 3(b)). Notably, we add a special textual token '$H\_$end' to mark the end of each hierarchy, where $H \in \{$curve, loop, face, sketch, extrusion$\}$. This is also different from Xu et al. (2022) where one-hot vectors are used as ending flags. We concatenate tokens of multiple curves to create the representation for a loop (the text with a green background in Fig. 3(b)). Then, we use a similar way to form representations of other hierarchies, including face and sketch (the texts with red and yellow backgrounds in Fig. 3(b)). Furthermore, an extrusion can also be represented using textual tokens, with the first one specifying its type (*e.g.*, add or cut) and the others denoting its numerical attributes (the text with a brown background in Fig. 3(b)). Finally, a complete CAD model is assembled by concatenating all the textual tokens

from its sketch-extrusions (the text with a cyan background in Fig. 3(b), please see the meaning of numbers in the text in the appendix). Consequently, we convert a CAD model into a structured text, enabling efficient processing and comprehension by LLMs. Besides, our FlexCAD shortens the overall token length compared to Xu et al. (2022). Moreover, this text is straightforward to parse and interpret, thereby facilitating the implementation of the below hierarchy-aware masking strategy.

## 3.2 FINE-TUNING LLMs WITH HIERARCHY-AWARE MASK PREDICTION

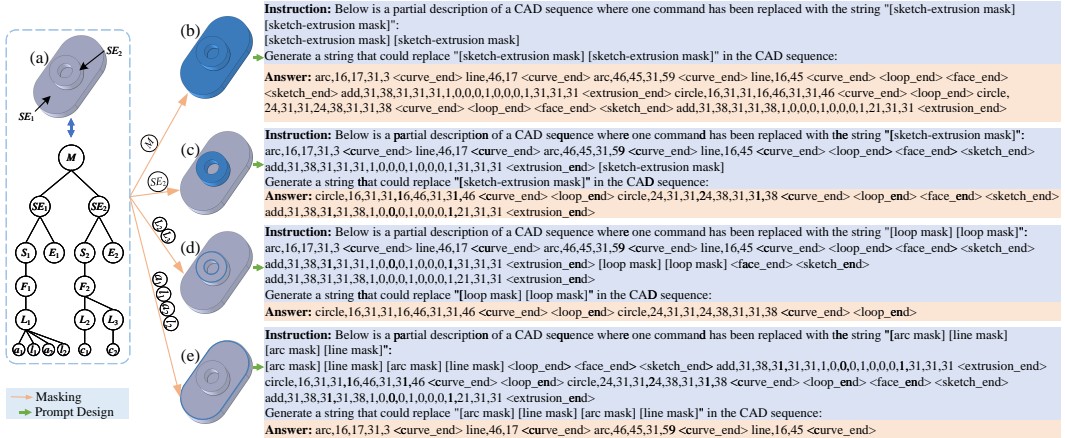

Figure 4:  (a) illustrates a CAD model and its structural diagram. (b), (c), (d) and (e) are four examples for prompt templates with the mask tokens designed to represent different construction hierarchies. The masked field for different hierarchies in the CAD model are highlighted in blue.

In the following, with the structured text representation (referred to as CAD text for simplicity), we introduce how to fine-tune LLMs to develop a unified model for various controllable CAD generation tasks. In general, during training, a hierarchy-aware field in the CAD text is replaced with a mask token. The field, which consists of a sequence of tokens, can be designed to reflect different CAD construction hierarchies. Next, LLMs are asked to predict the masked field (see Fig. 2(a)). To accomplish this, we design different prompt templates, where the mask tokens are designed to align with the corresponding hierarchies (see Fig. 4). During inference, given a CAD model, with a defined mask token, users can specify the part they want to modify (see Fig. 2(b)). Below, we further detail the design of prompt templates, the unified training and inference processes.

**Prompt Template Design with Hierarchy-aware Mask.** A prompt template includes an instruction with a special mask token replacing a hierarchy-aware field, and an answer containing the tokens of this field. Specifically, for the **CAD** level, we mask each internal sketch-extrusion with [sketch-extrusion mask]. In this case, other than the sketch-extrusion number, no information from the original CAD model is preserved. Fig. 4(b) shows an example. This allows us to freely generate CAD models with the expected number of sketch-extrusions during inference, facilitating the creation of CAD models with varying complexity. For the **sketch-extrusion**, **sketch** and **extrusion** levels, we replace the relevant field with [sketch-extrusion mask], [sketch mask], or [extrusion mask], respectively. Fig. 4(c) shows an example with one masked sketch-extrusion. For the **face** (**loop**) levels, given a face (loop), if it exclusively forms a sketch (face), we mask this face (loop) with [face mask] ([loop mask]). In cases where multiple faces (loops) belong to the same sketch (face), we use a corresponding number of mask tokens to mask them all at once. Fig. 4(d) illustrates an example, where two loops are replaced by two mask tokens. With this strategy, the model learns to generate faces (loops) with varying numbers as described in different instructions. For the **curve** level, all curves of the same loop are masked with their type indicated in the mask token (*i.e.*, line, arc or circle). Fig. 4(e) presents an example where four curves (arc-line-arc-line) belonging to the same loop are masked simultaneously. As the curve is the fundamental hierarchical level, the control of the topology and geometry of a sketch comes from it. Specifically, once trained and given a loop, by keeping its internal curve type and number unchanged, we can only modify the geometry. Alternatively, by varying the type or number of curves, we can alter the topology.

**Unified Training by Sampling Prompt Templates.** At each epoch, for a given CAD text, we uniformly sample a prompt template from the above seven hierarchies. The instruction in the template asks LLMs to predict the masked field autoregressively. Then, the cross-entropy (CE) loss between the prediction and the answer in the template is back-propagated to update the LLMs. To sum up, the advantage here is two-fold. First, by randomly choosing existing prompt templates at each epoch, we aim to establish a unified controllable generation model for various hierarchies. Second, beyond the existing prompt templates, we can incorporate new templates that support other tasks, such as unconditional generation. Notably, we fine-tune LLMs using LoRA Hu et al. (2022) which allows a few parameters to be trainable while keeping most parameters fixed. This allows us to leverage the advantages of large-scale models while accelerating model convergence Hu et al. (2022).

**Inference with User-defined Mask.** During inference, a CAD model is first converted to a CAD text, with a mask token replacing the part that needs modification. This masked CAD text is then input into the fine-tuned LLMs to produce predictions. After infilling the masked text with these predictions, FlexCAD can provide various CAD texts that can be rendered into diverse CAD models. Notably, users do not have to strictly adhere to the masking pattern defined in the training process. For example, although the prompt template, used in training, masks all the loops tied to a face simultaneously, this is not mandatory in the inference. Due to the strong generalization capability of LLMs, it is possible to only mask a single loop for local editing, as illustrated in Fig. 2(b).

## 4 EXPERIMENTS

### 4.1 EXPERIMENTAL SETUP

**Datasets.** For consistency with prior work Xu et al. (2022; 2023), we evaluate our FlexCAD on the DeepCAD Wu et al. (2021) dataset. This dataset comprises 178,238 sketch-and-extrusion sequences, divided randomly into training, validation, and test sets in a ratio of 90%-5%-5%. To ensure data quality, we follow SkexGen Xu et al. (2022) to remove duplicate and invalid sequences. Subsequently, we convert all resultant CAD sequences into texts, as mentioned in Sec. 3.1.

**Implementation Details.** We adopt the transformers Wolf et al. (2020) toolbox and select Llama-3-8B Meta (2024) as the base LLM, which achieves superior performance among open-source LLMs. For the 8B model, we use LoRA Hu et al. (2022) to fine-tune only 0.042% of their parameters, approximately 3.4 million. The LoRA rank and alpha are set to 8 and 32. The model is trained on four A6000 GPUs. we employ the AdamW optimizer Loshchilov & Hutter (2018), set the batch size to 32, use a cosine annealing learning rate of $5 \times 10^{-4}$, and train for 30 epochs. During the inference process, we set the sampling temperature $\tau$ and Top-p at 1.1 and 0.9, respectively.

**Metrics.** We adopt metrics consistent with previous methods Xu et al. (2022; 2023). Generally, the metrics Coverage (COV), Minimum Matching Distance (MMD) and Jensen-Shannon Divergence (JSD) measure generation diversity and quality on generated CAD models in comparison to the test set. See their detailed descriptions in Xu et al. (2023). Novel indicates the percentage of generated CAD models not present in the training set, while Unique represents the percentage of generated CAD models that appear only once within the generated set. Prediction Validity (PV) denotes the overall validity of predictions that can be rendered into 3D shapes rather than just 2D sketches or nothing. Realism denotes the realistic rate of generated CAD models compared to the training data, as assessed by human evaluators.

### 4.2 PERFORMANCE COMPARISION WITH EXISTING METHODS

**Baselines and Tasks.** We compare our FlexCAD with GPT-4o Achiam et al. (2023), one of the most powerful closed-source LLMs, and two state-of-the-art SEM-based baselines: SkexGen Xu et al. (2022) and Hnc-cad Xu et al. (2023). Since SkexGen and Hnc-cad cannot simultaneously control CAD generation across all hierarchies as ours, we choose the sketch-level and extrusion-level controllable generation tasks for comparison following the principles below. First, they are common tasks that can be handled by each baseline. Second, there are official implementations of baselines for these tasks. For GPT-4o, Hnc-cad and FlexCAD, given a CAD model from the test set, we randomly mask either a sketch or an extrusion and predict the corresponding masked field. Notably, despite our best efforts, SkexGen still has slightly different task settings compared to the

Table 1: Performance comparison on the DeepCAD test set. GPT-4o is enhanced with few-shot in-context learning. Specifically, each prompt comprises five exemplars randomly chosen from the training set. These exemplars include instructions and answers, following the format shown in Fig. 4. Moreover, we obtain the performance values of SkexGen and Hnc-cad based on their official codes. Best performances are in **bold**, and the second-bests are marked by *.

| Method | Sketch-level | | | | | | | Extrusion-level | | | | | | |
|---|---|---|---|---|---|---|---|---|---|---|---|---|---|---|
| | COV↑ | MMD↓ | JSD↓ | Novel↑ | Unique↑ | PV↑ | Realism↑ | COV↑ | MMD↓ | JSD↓ | Novel↑ | Unique↑ | PV↑ | Realism↑ |
| GPT-4o | 58.2% | 1.34 | 1.43 | 69.7% | 72.8% | 62.3% | 23.2% | 53.3% | 1.42 | 2.14 | 58.6% | 65.3% | 48.8% | 19.7% |
| SkexGen | 60.6% | 1.27 | 1.51 | 90.7%* | **93.5%** | 68.7% | 34.8% | 63.6% | 1.23 | 1.44 | **89.3%** | 89.1%* | 76.1% | 35.2% |
| Hnc-cad | 62.4%* | 1.21* | 1.07* | 87.6% | 92.1% | 72.6%* | 36.3%* | 65.6%* | 1.25* | 1.38* | 86.2% | 87.8% | 79.7%* | 38.0% * |
| Ours | **65.6%** | **1.19** | **0.82** | **92.1%** | 92.6%* | **93.4%** | **39.6%** | **68.5%** | **1.19** | **1.32** | 87.6%* | **90.4%** | **93.3%** | **42.1%** |

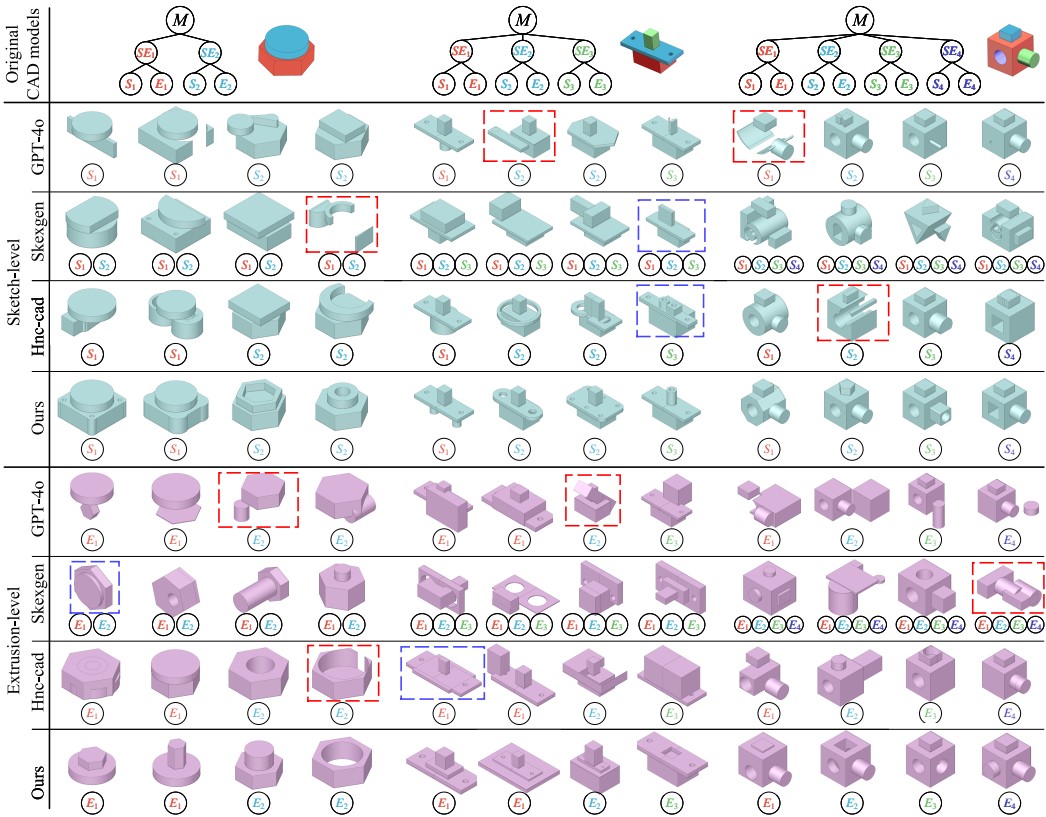

Figure 5: Qualitative comparison results for four methods. The first row displays three original CAD models, where the color of each sketch-extrusion aligns with that in the corresponding structural diagrams. In the following rows, given a CAD model, we randomly select its four newly predicted models for each method. The marks below the predictions are the corresponding masked and modified sketches or extrusions. The red boxes illustrate some of the most unrealistic examples. The blue boxes indicate some of the most obvious cases, where multiple fields simultaneously change in the same CAD model, rather than just the expected masked field.

above three methods. Specifically, when there are multiple sketches (or extrusions) in a CAD model, SkexGen changes all the sketches (or extrusions) instead of the specified one.

**Quantitative Comparison Results.** We randomly selected 1k CAD models from the test set. For every method, we generated 10 predictions for each model, resulting in a total of 10k CAD models. For the metrics COV, MMD, and JSD that require a subset of ground truths, we sampled 3k CAD models from the test set. The average scores across three runs are presented in Table 1. Without fine-tuning, GPT-4o performs poorly. On the other hand, our FlexCAD outperforms the baselines

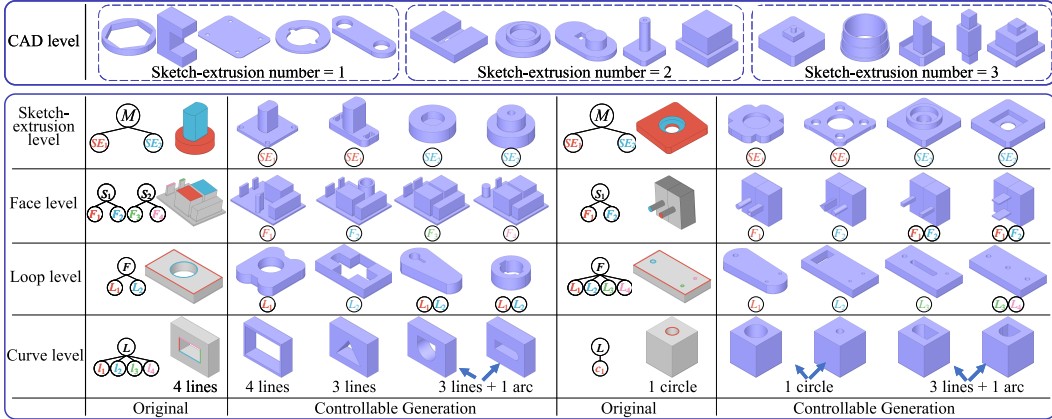

Figure 6: Our FlexCAD achieves controllable generation across different hierarchies, as introduced in 3.2. For the CAD level, we produce CAD models aligning with the required sketch-extrusion number. For the sketch-extrusion, face and loop levels, the left side of each sub-figure shows an original CAD model along with its local structural diagram. The color of each highlighted field matches that in the diagrams. The right side shows the predictions with only the masked part being masked and edited. And the masked part is marked below the predictions. Similarly, for the curve level, below the predictions are user-defined curve type and number. Best viewed in color.

on nearly all evaluation metrics, demonstrating significant superiority in generation quality and controllability. Particularly, FlexCAD achieves the most notable improvement on PV, reaching up to 20.8%-31.1% and 13.6%-44.5% in terms of sketch-level and extrusion-level controllable generation.

**Qualitative Comparison Results.** To illustrate the performance intuitively, we randomly selected three CAD models from the test set. As shown in Fig. 5, the results clearly illustrate that FlexCAD greatly enhances the quality and controllability of CAD models compared to other competitors. Specifically, we tend to generate well-structured CAD models that closely resemble real-world examples, contrasting with unrealistic models like the red boxes shown in Fig. 5. On the other hand, as shown in the blue boxes in Fig. 5, SkexGen cannot specify which sketch or extrusion to modify, and Hnc-cad cannot preserve the integrity of the unmasked elements, even when recovering from the same codes. In contrast, we can mask any sketch or extrusion, ensuring that only the masked sketch or extrusion is modified while the remaining elements stay unchanged. This further confirms the effectiveness and superior controllability of our FlexCAD.

**Human Evaluation.** To evaluate Realism, for each method, seven crowd workers were shown 950 pairs of images from the generated data and the training data, following Xu et al. (2023). They were asked to judge which of the two was more realistic. As shown in Table 1, for our FlexCAD, 39.6% and 42.1% of the generated models are more realistic. These rates are the highest among all methods, further demonstrating the advantage of our FlexCAD.

### 4.3 ENABLING MORE CONTROLLABLE GENERATION TASKS

In addition to the sketch and extrusion levels, our FlexCAD achieves controllable generation in other hierarchies, including the CAD, sketch-extrusion, face, loop, and curve levels. We provide some examples in Fig. 6, where users can modify the specific fields of a CAD model according to their intent. Additionally, we present detailed qualitative results across these hierarchies in Table 5 in the appendix. All the results together illustrate the effectiveness of our FlexCAD.

Furthermore, our FlexCAD can achieve iterative editing. For example, as shown in Fig. 7(a), starting from a simple CAD model, we iteratively edit it within a newly generated sketch-extrusion until the sketch-extrusion aligns closely with user requirements. On the other hand, as shown in Fig. 7(b), given a complex CAD model, we continuously adjust diverse hierarchy elements within different sketch-extrusions until they progressively match user specifications.

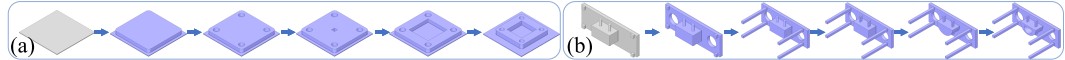

Figure 7: Two examples of iterative editing. (a) Based on a simple CAD model, a new sketch-extrusion is generated by adding a [sketch-extrusion mask] at the end of the CAD text. Similarly, four peripheral internal loops are created. Then, a central quadrilateral loop is added and its geometry is altered. Finally, the new extrusion is adjusted to better match the original model. (b) Based on a complex CAD model, modifications can be progressively applied at various sketch-extrusions with the loop-extrusion-face-curve-loop level controllable generation to ultimately meet user needs.

Table 2: Ablation studies for fine-tuning LLMs with different settings. Pre-trained denotes the initial pre-trained weights. Full and LoRA indicate full and a few parameters are trainable, respectively. Transformer-4M is a small transformer-based Vaswani et al. (2017) language model. Its total number of trainable parameters is comparable to that of our model with LoRA. Llama-3-8B-From-Scratch and Llama-3-8B-Full denote training full parameters without and with the initial pre-trained weights, respectively. Llama-3-8B-Instruct is an instruction-tuned model in an 8B size Meta (2024). For Llama-3-70B Meta (2024), we fine-tune only 0.023% of its parameters, around 16.3 million. Best performances are in **bold** and the second-bests are marked by ∗.

| Model | COV↑ | MMD↓ | JSD↓ | Novel↑ | Unique↑ | PV↑ |
|---|---|---|---|---|---|---|
| Transformer-4M (w/o Pre-trained, Full) | 59.4% | 1.37 | 1.02 | 85.8% | 86.9% | 80.2% |
| Llama-3-8B-From-Scratch (w/o Pre-trained, Full) | 63.0% | 1.23 | 0.91 | 89.7% | 90.2% | 89.5% |
| Llama-3-8B-Full (w/ Pre-trained, Full) | 66.4%∗ | 1.20 | 0.85 | 92.6%∗ | 91.0% | 91.7% |
| Llama-3-8B-Instruct (w/ Pre-trained, LoRA) | 65.3% | 1.22 | 0.89 | 91.4% | 92.1%∗ | 90.5% |
| Llama-3-8B (w/ Pre-trained, LoRA, ours) | 65.6% | 1.19∗ | **0.82** | 92.1% | **92.6%** | 93.4%∗ |
| Llama-3-70B (w/ Pre-trained, LoRA) | **68.2%** | **1.13** | 0.84∗ | **93.0%** | 91.8% | **94.6%** |

Table 3: Effectiveness analysis of the hierarchy-aware masking strategy and unified training. Random Masking denotes randomly masking 15%-50% continuous tokens within each CAD text, instead of the hierarchy-aware field. w/o Hierarchy-specific Tokens indicates that when masking, we utilize the generic token [mask], rather than employing hierarchy-specific mask tokens, such as [face mask], [loop mask] and etc. w/o Unified Training represents that we solely train a single task, *i.e.*, the sketch-level controllable generation. Best performances are in **bold**.

| Model | COV↑ | MMD↓ | JSD↓ | Novel↑ | Unique↑ | PV↑ |
|---|---|---|---|---|---|---|
| Random Masking | 63.0% | 1.25 | 1.02 | 88.2% | 91.5% | 90.6% |
| w/o Hierarchy-specific Tokens | 63.7% | 1.20 | 0.95 | 90.8% | 91.7% | 91.5% |
| w/o Unified Training | 64.3% | **1.17** | 0.89 | 91.6% | 90.9% | 92.2% |
| Ours | **65.6%** | 1.19 | **0.82** | **92.1%** | **92.6%** | **93.4%** |

## 4.4 ABLATION STUDIES

We conduct several ablation studies evaluated on the sketch-level controllable generation as mentioned in Sec. 4.2, unless otherwise stated.

**Settings of LLMs.** As shown in Table 2, without the pre-trained weights, both Transformer-4M and Llama-3-8B-From-Scratch achieve the lowest performances. This indicates that the pre-trained weights in LLMs contain valuable knowledge, which contributes to the performance gains. Llama-3-8B-Full achieves a performance similar to that of our model. However, it requires 80 hours to reach model convergence, compared to just 20 hours for ours. This highlights the effectiveness of the LoRA strategy Hu et al. (2022). The performance of Llama-3-8B-Instruct is slightly lower than that of our model. Conversely, as the model scale increases significantly, Llama-3-70B achieves the best performance but is more time-consuming and costly (See Sec. A.6 in the appendix).

**Effectiveness of Hierarchy-Aware Masking.** As shown in Table 3, Random Masking exhibits the lowest performance, underscoring the effectiveness of masking the hierarchy-aware field. Furthermore, the absence of hierarchy-specific tokens leads to a performance decline. In essence, these

tokens potentially assist LLMs in more accurately determining the level they are generating, consequently enhancing overall controllable generation performance.

**Effectiveness of Unified Training.** As depicted in Table 3, without unified training, there is a slight decrease in performance. In other words, when simultaneously training multiple controllable generation tasks across various levels, the inter-task knowledge contributes to performance gains.

## 5 CONCLUSION

In this paper, we introduce a unified, versatile and user-friendly model, termed FlexCAD, which is particularly designed for controlling CAD generation across all hierarchies. To the best of our knowledge, we are the first to utilize LLMs for controllable CAD generation. Specifically, we convert each CAD model into a brief and structured text and propose hierarchy-aware masking for fine-tuning. Our FlexCAD is simple yet highly effective. Thorough qualitative and quantitative assessments conducted on public benchmarks confirm its effectiveness across all hierarchies.

## ETHICS STATEMENT

The data used in this work is tailored for creating and modifying CAD models. Due to its specialized nature, the misuse risk is naturally minimized, ensuring that the developed methods primarily benefit design and engineering tasks. In this work, we have invited crowd workers to evaluate the quality of generated CAD models. We conducted this work in accordance with ethical guidelines to ensure that participants were treated fairly, respectfully, and safely throughout the process. We took steps to protect the privacy of crowd workers by not collecting personally identifiable information. The data annotated by the crowd workers was used only for research purpose related to improving CAD generating techniques.

## ACKNOWLEDGEMENT

In this work, Zhanwei Zhang, Wenxiao Wang and Deng Cai were supported in part by The National Nature Science Foundation of China (Grant No: 62303406, 62273302, 62036009, 61936006), in part by Zhiyuan Laboratory (NO. ZYL2024022b).

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

# Appendix

Considering the space limitation of the main paper, we provided more results and discussion in this appendix, which is organized as follows:

## A  ADDITIONAL IMPLEMENTATION DETAILS AND ANALYSIS

### A.1  MEANING OF NUMBERS IN THE CAD TEXT.

Table 4: Effectiveness analysis of circle representation. To denote a circle, $\mathrm{Center\&Radius}$ utilizes the center coordinates along with the radius, while $\mathrm{Diameter}$ uses two uniformly distributed points on the circumference that collectively define the diameter. Best performances are in **bold**.

| Model | COV↑ | MMD↓ | JSD↓ | Novel↑ | Unique↑ | PV↑ |
|---|---|---|---|---|---|---|
| Center&Radius | 63.2% | 1.21 | 0.87 | 90.2% | 90.9% | 89.7% |
| Diameter | **66.4%** | 1.20 | 0.87 | 90.7% | 91.5% | 90.3% |
| Four points (Ours) | 65.6% | **1.19** | **0.82** | **92.1%** | **92.6%** | **93.4%** |

In this part, we explain the meaning of numbers in the CAD text, which builds upon SkexGen Xu et al. (2022). In sketches, two (three) points can actually form a line (arc). In our work, a line, an arc, and a circle are represented by one, two, and four points, respectively. Each point is denoted as its $x$ and $y$ coordinates. Here, the second (third) point of a line (arc) is determined by the first point of the subsequent curve (or the first curve when a loop is closed). Four points of a circle are uniformly distributed along the circumference. We implement two variants of circle representation. As shown in Table 4, when evaluated on the sketch-level controllable generation task, our FlexCAD displays robustness across different circle representations, with $\mathrm{Four\ points}$ showing a slight edge.

Each extrusion operation is represented by 18 parameters: BVVTTTRRRRRRRRRSOO.
- B represents one of the three Boolean operations: add, cut or intersect. It occupies 1 parameter.
- V indicates the displacements of the top and the bottom planes from the reference plane in which a sketch is extruded to form a solid. It occupies 2 parameters.
- T represents the 3D translation applied to the extruded solid. It occupies 3 parameters.
- R represents the 3D rotation of the extrusion direction. It occupies 9 parameters.
- S represents the uniform scaling factor. It occupies 1 parameter.
- O represents the center of scaling as a 2D coordinate. It occupies 2 parameters.

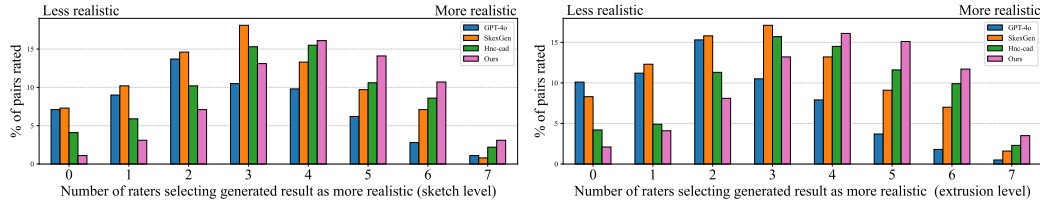

Figure 8: Distribution of realism scores from seven human evaluators. These scores are derived by comparing the generated CAD models produced by the four methods with the training samples.

## A.2 DETAILED RESULTS FOR HUMAN EVALUATION

We present the detailed distribution of the Realism scores as mentioned in Table 1. As shown in the Fig. 8, the distributions for GPT-4o, SkexGen, and Hnc-cad are skewed towards the 'less realistic' end. Conversely, our FlexCAD demonstrates a primarily symmetric distribution, suggesting that the crowd workers struggle to differentiate between the generated models and the training set.

## A.3 QUANTITATIVE RESULTS FOR OTHER HIERARCHIES

Table 5: Quantitative results for controllable generation across other hierarchies. The detailed evaluation setting is the same as that of Table 1.

| Hierarchy | COV↑ | MMD↓ | JSD↓ | Novel↑ | Unique↑ | PV↑ |
|---|---|---|---|---|---|---|
| CAD level | 67.2% | 1.14 | 0.77 | 92.6% | 93.2% | 91.8% |
| Sketch-extrusion level | 65.3% | 1.21 | 0.80 | 90.3% | 89.7% | 90.5% |
| Face level | 62.9% | 1.18 | 0.84 | 91.1% | 90.9% | 93.2% |
| Loop level | 63.4% | 1.15 | 0.81 | 88.3% | 85.7% | 90.5% |
| Curve level | 59.1% | 1.20 | 0.79 | 89.7% | 91.5% | 90.2% |

In this section, we report the quantitative results across other hierarchies, including CAD, sketch-extrusion, face, loop and curve levels. By combining the data from Table 1 and Table 5, we observe that there is not a significant difference in performance across all hierarchies. These results together illustrate the effectiveness of our FlexCAD across all hierarchies.

## A.4 SENSITIVITY ANALYSIS OF KEY HYPER-PARAMETERS IN SAMPLING

Table 6: Effectiveness analysis of key hyper-parameters, including the sampling temperature $\tau$ and Top-p. Best performances are in **bold** and the second-bests are marked by *.

| Model | COV↑ | MMD↓ | JSD↓ | Novel↑ | Unique↑ | PV↑ |
|---|---|---|---|---|---|---|
| $\tau = 0.9$ | 62.7% | 1.20 | 0.84 | 92.0%* | 90.3% | **96.7%** |
| $\tau = 1.1$ | 65.6%* | 1.19* | 0.82* | **92.1%** | 92.6%* | 93.4%* |
| $\tau = 1.3$ | **65.8%** | **1.12** | **0.78** | 91.9% | **94.5%** | 86.8% |
| Top-p $= 0.8$ | 62.9% | 1.23 | 0.90 | 91.7% | 87.8% | **95.2%** |
| Top-p $= 0.9$ | 65.6%* | 1.19* | 0.82* | 92.1%* | 92.6%* | 93.4%* |
| Top-p $= 1.0$ | **68.3%** | **1.13** | **0.77** | **93.3%** | **95.7%** | 89.0% |

In this part, we perform a sensitivity analysis on key hyperparameters in inference, including the sampling temperature $\tau$ and Top-p. As shown in Table 6, as either $\tau$ or Top-p increases, the performance of the first five metrics exhibits improvement, whereas the performance of the last deteriorates. Essentially, higher values of $\tau$ or Top-p lead to predictions that are more random and varied, while the overall prediction validity PV declines. In our experiments, we made a trade-off by selecting the values of $\tau$ and Top-p to guarantee that the PV value remains above 90%.

## A.5 UNCONDITIONAL GENERATION TASK

Table 7: Performance comparison for the unconditional generation task. Each method generates 10k CAD models, which are then compared with a randomly selected subset of 2.5k ground truth models from the test set. For the baselines, other than the metric PV, we derive the values of other metrics from the original paper Xu et al. (2023), while the values of PV are obtained based on their official codes. Best performances are in **bold**, and the second-bests are marked by *.

| Model | COV↑ | MMD↓ | JSD↓ | Novel↑ | Unique↑ | PV↑ | Realism |
|---|---|---|---|---|---|---|---|
| SkexGen | 84.7% | 1.02 | 0.90* | 99.1%* | **99.8%** | 74.2% | 46.9% |
| Hnc-cad | 87.7%* | 0.96* | **0.68** | 93.9% | 99.7%* | 77.4%* | 49.2% |
| Ours | **89.2%** | **0.91** | 1.53 | **99.3%** | 96.9% | **90.5%** | 51.5% |

Our FlexCAD can easily achieve unconditional CAD generation by simply adding a prompt template during training. Specifically, given a CAD text, the instruction in the prompt template can be as concise as '*Below is a description of a CAD sequence:*', while the corresponding answer is the whole CAD text. The quantitative and qualitative results shown in Table 7 and Figure 17 verify the effectiveness of our FlexCAD in unconditional CAD generation. Notably, as shown in Table 7, our JSD exhibits the poorest performance. However, upon adjusting the sampling temperature $\tau$ or Top-p to maintain the PV value at around 80%, the JSD value enhances significantly to 0.78.

## A.6 LIMITATIONS AND FUTURE WORK

**Inference time.** We measure the inference time on one A6000 GPU (with a batch size of 1) for the extrusion-level generation, averaging over 1,000 runs. The inference time of our FlexCAD (based on Llama-3-8B) is slightly higher than that of SkexGen Xu et al. (2022) and Hnc-cad Xu et al. (2022), at 0.56 seconds compared to 0.15 seconds and 0.38 seconds, respectively. Notably, we trained and tested Llama-3-70B using four A100 GPUs (with a batch size of 1 per GPU), yet the average inference time is still close to 3 seconds. Although LLMs demonstrate promising performance, they generally lack efficiency. while the task of controllable CAD generation is not particularly demanding in terms of real-time inference requirements, the slight increase in inference time of our FlexCAD (based on Llama-3-8B) is acceptable given the promising performance.

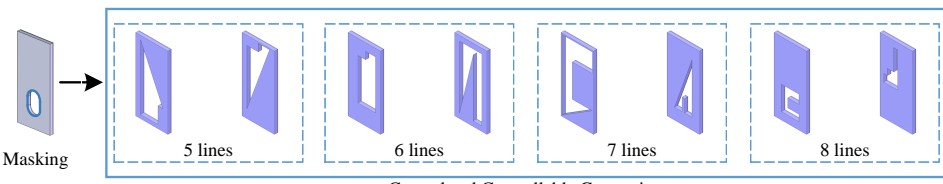

Figure 9: Failure cases. The generated CAD models display irregular polygonal shapes.

**Failure cases.** Despite the significant progress, our FlexCAD sometimes generates hallucinations. For example, when we implement curve-level controllable generation by replacing the original loops with 5, 6, 7, or 8 lines, our FlexCAD tends to generate plausible CAD texts. The CAD models produced from these texts typically exhibit irregular polygonal shapes, as illustrated in Fig. 9. Interestingly, when the modifications are limited to 4 lines, the generated predictions tend to be more structurally regular, as illustrated in Fig. 16. We select loops composed of 4, 5, 6, 7, or 8 lines in the training set for analysis. The ratio is approximately 77.9%-6.2%-9.5%-1.7%-4.7%. Consequently, the fundamental reason behind these hallucinations may stem from data imbalance, which could potentially be mitigated by collecting additional data.

**Other limitations and future work.** In this paper, we fine-tune LLMs to achieve controllable CAD generation. However, several unexplored domains warrant investigation. One such domain involves determining how LLMs can be effectively utilized for text-based generation tasks, such as providing a descriptive text like '*Create a desk.*' to generate a corresponding CAD model. Since LLMs possess robust capabilities in controllable CAD generation and text comprehension, in future work, we aim to build a more advanced multi-modal LLM tailored for the text-based CAD generation task.

**Instruction:** Below is a partial description of a CAD sequence where one command has been replaced with the string "[sketch-extrusion mask]":
[sketch-extrusion mask]
Generate a string that could replace "[sketch-extrusion mask]" in the CAD sequence:
**Predictions:**

(1) arc,23,27,9,31 <curve_end> arc,23,35,25,38 <curve_end> arc,28,39,31,53 <curve_end> arc,34,39,37,38 <curve_end> arc,39,35,53,31 <curve_end> arc,39,27,37,24 <curve_end> arc,34,23,31,9 <curve_end> arc,28,23,25,24 <curve_end> <loop_end> circle,10,31,16,25,22,31,16,37 <curve_end> <loop_end> circle,25,31,31,25,37,31,31,37 <curve_end> <loop_end> circle,25,16,31,10,37,16,31,22 <curve_end> <loop_end> circle,25,46,31,40,37,46,31,52 <curve_end> <loop_end> circle,40,31,46,25,52,31,46,37 <curve_end> <loop_end> <face_end> <sketch_end> add,31,63,31,31,31,1,0,0,0,0,1,0,-1,0,47,31,31 <extrusion_end>

(2) line,9,9 <curve_end> line,53,9 <curve_end> line,53,53 <curve_end> line,9,53 <curve_end> <loop_end> line,14,17 <curve_end> line,20,17 <curve_end> line,20,24 <curve_end> line,14,24 <curve_end> <loop_end> line,14,38 <curve_end> line,20,38 <curve_end> line,20,45 <curve_end> line,14,45 <curve_end> <loop_end> line,42,17 <curve_end> line,48,17 <curve_end> line,48,24 <curve_end> line,42,24 <curve_end> <loop_end> line,42,38 <curve_end> line,48,45 <curve_end> line,48,45 <curve_end> <loop_end> <face_end> <sketch_end> add,31,52,31,31,31,1,0,0,0,1,0,0,0,1,62,31,31 <extrusion_end>

(3) circle,9,31,31,9,53,31,31,53 <curve_end> <loop_end> circle,15,31,18,27,22,31,18,35 <curve_end> <loop_end> circle,27,18,31,14,35,18,31,22 <curve_end> <loop_end> circle,27,44,31,40,35,44,31,48 <curve_end> <loop_end> circle,40,31,44,27,47,31,44,35 <curve_end> <loop_end> <face_end> <sketch_end> add,31,34,31,31,31,1,0,0,0,0,1,0,-1,0,62,31,31 <extrusion_end>

**Text-to-CAD Rendering**

Sketch-extrusion number = 1 ⟶

**Instruction:** Below is a partial description of a CAD sequence where one command has been replaced with the string "[sketch-extrusion mask]":
[sketch-extrusion mask] [sketch-extrusion mask]
Generate a string that could replace "[sketch-extrusion mask] [sketch-extrusion mask]" in the CAD sequence:
**Predictions:**

(4) circle,9,31,31,9,53,31,31,53 <curve_end> <loop_end> <face_end> <sketch_end> add,31,39,31,31,31,1,0,0,0,1,0,0,0,1,62,31,31 <extrusion_end> arc,9,16,16,16 <curve_end> line,16,9 <curve_end> arc,46,9,46,16 <curve_end> line,53,16 <curve_end> arc,53,46,46,46 <curve_end> line,46,53 <curve_end> arc,16,53,16,46 <curve_end> line,9,46 <curve_end> <loop_end> circle,19,31,31,19,43,31,31,43 <curve_end> <loop_end> <face_end> <sketch_end> add,31,46,31,31,39,1,0,0,0,1,0,0,0,1,47,31,31 <extrusion_end>

(5) circle,9,31,31,9,53,31,31,53 <curve_end> <loop_end> circle,10,31,12,30,13,31,12,32 <curve_end> <loop_end> circle,17,20,18,19,19,20,18,21 <curve_end> <loop_end> circle,17,42,18,41,19,42,18,43 <curve_end> <loop_end> circle,30,12,31,10,32,12,31,13 <curve_end> <loop_end> circle,30,50,31,49,32,50,31,52 <curve_end> <loop_end> circle,43,20,44,19,45,20,44,21 <curve_end> <loop_end> circle,43,42,44,41,45,42,44,43 <curve_end> <loop_end> circle,49,31,50,30,52,31,50,32 <curve_end> <loop_end> <face_end> <sketch_end> add,31,34,31,31,31,1,0,0,0,1,0,0,0,1,31,31,31 <extrusion_end> circle,9,31,31,9,53,31,31,53 <curve_end> <loop_end> <face_end> <sketch_end> add,31,63,31,31,34,1,0,0,0,1,0,0,0,1,6,31,31 <extrusion_end>

(6) line,6,11 <curve_end> line,56,11 <curve_end> line,56,51 <curve_end> line,6,51 <curve_end> <loop_end> <face_end> <sketch_end> add,31,36,31,31,31,1,0,0,0,0,1,0,-1,0,47,30,37 <extrusion_end> line,6,51 <curve_end> line,6,11 <curve_end> line,56,11 <curve_end> line,56,51 <curve_end> <loop_end> <face_end> <sketch_end> cut,19,31,30,31,36,-1,0,0,0,0,1,0,1,0,31,31,31 <extrusion_end>

**Text-to-CAD Rendering**

Sketch-extrusion number = 2 ⟶

**Instruction:** Below is a partial description of a CAD sequence where one command has been replaced with the string "[sketch-extrusion mask] [sketch-extrusion mask]":
[sketch-extrusion mask] [sketch-extrusion mask] [sketch-extrusion mask]
Generate a string that could replace "[sketch-extrusion mask] [sketch-extrusion mask] [sketch-extrusion mask]" in the CAD sequence:
**Predictions:**

(7) circle,9,31,31,9,53,31,31,53 <curve_end> <loop_end> <face_end> <sketch_end> add,31,46,31,31,31,1,0,0,0,0,1,0,-1,0,19,31,31 <extrusion_end> circle,9,31,31,9,53,31,31,53 <curve_end> <loop_end> circle,25,31,31,25,37,31,31,37 <curve_end> <loop_end> <face_end> <sketch_end> add,31,43,31,31,31,-1,0,0,0,0,1,0,1,0,13,31,31 <extrusion_end> circle,9,31,31,9,53,31,31,53 <curve_end> <loop_end> <face_end> <sketch_end> add,31,55,31,31,31,-1,0,0,0,0,1,0,1,0,3,31,31 <extrusion_end>

(8) line,4,15 <curve_end> line,58,15 <curve_end> line,58,47 <curve_end> line,4,47 <curve_end> <loop_end> <face_end> <sketch_end> add,31,44,31,31,31,1,0,0,0,0,1,0,-1,0,25,48,41 <extrusion_end> line,5,14 <curve_end> line,57,14 <curve_end> line,57,48 <curve_end> line,5,48 <curve_end> <loop_end> <face_end> <sketch_end> cut,18,31,46,24,49,1,0,0,0,1,0,0,0,1,11,41,29 <extrusion_end> arc,9,12,10,10 <curve_end> line,12,9 <curve_end> arc,50,9,52,10 <curve_end> line,53,12 <curve_end> arc,53,50,52,52 <curve_end> line,50,53 <curve_end> arc,12,53,10,52 <curve_end> line,9,50 <curve_end> <loop_end> <face_end> <sketch_end> cut,26,31,46,24,49,1,0,0,0,1,0,0,0,1,11,41,33 <extrusion_end>

(9) line,5,14 <curve_end> line,57,14 <curve_end> line,57,31 <curve_end> line,14,31 <curve_end> line,14,48 <curve_end> line,5,48 <curve_end> <loop_end> <face_end> <sketch_end> add,21,41,31,31,31,1,0,0,0,0,1,0,-1,0,26,48,42 <extrusion_end> line,9,9 <curve_end> line,53,9 <curve_end> line,53,53 <curve_end> line,9,53 <curve_end> <loop_end> <face_end> <sketch_end> cut,26,36,31,31,31,1,0,0,0,0,1,0,-1,0,10,60,37 <extrusion_end> arc,12,22,32,6 <curve_end> line,50,26 <curve_end> line,38,56 <curve_end> line,32,56 <curve_end> <loop_end> <face_end> <sketch_end> cut,26,36,31,31,31,1,0,0,0,0,1,0,-1,0,12,42,42 <extrusion_end>

**Text-to-CAD Rendering**

Sketch-extrusion number = 3 ⟶

Figure 10: CAD level generation. Here, users can produce CAD models that match the required sketch-extrusion number. The marks of predictions align with those of the images (the same below).

**Instruction:** Below is a partial description of a CAD sequence where one command has been replaced with the string "[sketch-extrusion mask]":
arc,12,14,31,6 <curve_end> line,50,14 <curve_end> arc,50,48,31,56 <curve_end> line,12,48 <curve_end> <loop_end> <face_end> <sketch_end>
add,31,55,31,31,31,1,0,0,0,1,0,0,0,1,52,31,31 <extrusion_end> [sketch-extrusion mask]
Generate a string that could replace " [sketch-extrusion mask]" in the CAD sequence:

**Predictions:**

(1) arc,15,15,31,4 <curve_end> line,47,15 <curve_end> arc,47,47,31,58 <curve_end> line,15,47 <curve_end> <loop_end> <face_end> <sketch_end> add,31,63,31,31,31,1,0,0,0,1,0,0,0,1,33,31,31 <extrusion_end>

(2) arc,8,14,31,10 <curve_end> line,54,14 <curve_end> arc,54,48,31,52 <curve_end> line,8,48 <curve_end> <loop_end> circle,20,31,31,20,42,31,31,42 <curve_end> <loop_end> <face_end> <sketch_end> add,31,63,31,31,31,1,0,0,0,1,0,0,0,1,23,31,31 <extrusion_end>

(3) line,10,19 <curve_end> line,31,7 <curve_end> line,52,19 <curve_end> line,52,43 <curve_end> line,31,55 <curve_end> line,10,43 <curve_end> <loop_end> <face_end> <sketch_end> cut,22,31,31,31,55,1,0,0,0,1,0,0,0,1,22,31,31 <extrusion_end>

(4) arc,15,13,31,4 <curve_end> line,47,13 <curve_end> arc,47,49,31,58 <curve_end> line,15,49 <curve_end> <loop_end> <face_end> <sketch_end> cut,25,31,31,31,55,1,0,0,0,1,0,0,0,1,30,31,31 <extrusion_end>

(5) arc,10,19,31,7 <curve_end> line,52,19 <curve_end> arc,52,43,31,55 <curve_end> line,10,43 <curve_end> <loop_end> circle,19,31,31,19,43,31,31,43 <curve_end> <loop_end> <face_end> <sketch_end> add,31,62,31,31,55,1,0,0,0,1,0,0,0,1,36,31,31 <extrusion_end>

(6) circle,2,31,13,20,24,31,13,42 <curve_end> <loop_end> <face_end> circle,38,31,49,20,60,31,49,42 <curve_end> <loop_end> <face_end> <sketch_end> cut,19,31,31,31,55,1,0,0,0,1,0,0,0,1,33,31,31 <extrusion_end>

(7) circle,9,31,31,9,53,31,31,53 <curve_end> <loop_end> <face_end> <sketch_end> cut,19,43,31,31,31,1,0,0,0,0,1,0,-1,0,14,31,55 <extrusion_end>

(8) arc,6,18,7,13 <curve_end> line,11,12 <curve_end> arc,23,25,23,27 <curve_end> line,25,29 <curve_end> arc,36,29,38,27 <curve_end> line,38,25 <curve_end> arc,51,12,55,13 <curve_end> line,56,18 <curve_end> arc,56,44,55,49 <curve_end> line,51,50 <curve_end> arc,38,37,38,35 <curve_end> line,36,33 <curve_end> arc,25,33,23,35 <curve_end> line,23,37 <curve_end> arc,11,50,7,49 <curve_end> line,6,44 <curve_end> <loop_end> <face_end> <sketch_end> add,31,33,31,31,55,1,0,0,0,1,0,0,0,1,22,31,31 <extrusion_end>

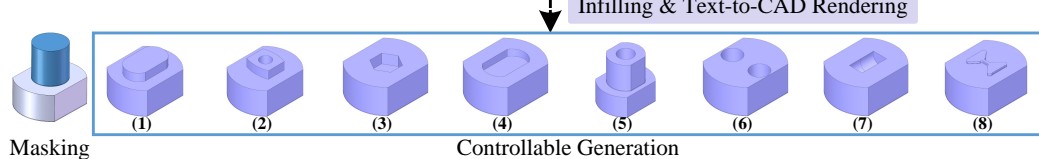

Masking        Infilling & Text-to-CAD Rendering

(1) (2) (3) (4) (5) (6) (7) (8)

Controllable Generation

**Instruction:** Below is a partial description of a CAD sequence where one command has been replaced with the string "[sketch-extrusion mask]":
[sketch-extrusion mask] circle,9,31,31,9,53,31,31,53 <curve_end> <loop_end> <face_end> <sketch_end>
add,31,63,31,31,55,1,0,0,0,1,0,0,0,1,31,31,31 <extrusion_end>
Generate a string that could replace " [sketch-extrusion mask]" in the CAD sequence:

**Predictions:**

(9) arc,9,12,10,10 <curve_end> line,12,9 <curve_end> arc,50,9,52,10 <curve_end> line,53,12 <curve_end> arc,53,50,52,52 <curve_end> line,50,53 <curve_end> arc,12,53,10,52 <curve_end> line,9,50 <curve_end> <loop_end> circle,11,12,12,11,14,12,12,14 <curve_end> <loop_end> circle,11,50,12,48,14,50,12,51 <curve_end> <loop_end> circle,48,12,50,11,51,12,50,14 <curve_end> <loop_end> circle,48,50,50,48,51,50,50,51 <curve_end> <loop_end> <face_end> <sketch_end> add,31,63,31,31,31,1,0,0,0,1,0,0,0,1,53,31,31 <extrusion_end>

(10) line,8,31 <curve_end> line,20,9 <curve_end> line,42,9 <curve_end> line,54,31 <curve_end> line,42,53 <curve_end> line,20,53 <curve_end> <loop_end> <face_end> <sketch_end> add,31,63,31,31,31,1,0,0,0,1,0,0,0,1,54,31,31 <extrusion_end>

(11) circle,9,31,31,9,53,31,31,53 <curve_end> <loop_end> circle,22,31,31,22,40,31,31,40 <curve_end> <loop_end> <face_end> <sketch_end> add,31,63,31,31,31,1,0,0,0,1,0,0,0,1,58,31,31 <extrusion_end>

(12) line,9,9 <curve_end> line,9,53 <curve_end> line,53,53 <curve_end> line,53,9 <curve_end> <loop_end> <face_end> <sketch_end> add,31,55,31,31,31,1,0,0,0,1,0,0,0,1,62,31,31 <extrusion_end>

(13) arc,8,30,12,13 <curve_end> line,29,9 <curve_end> arc,33,9,50,13 <curve_end> line,54,30 <curve_end> arc,54,32,50,49 <curve_end> line,33,53 <curve_end> arc,29,53,12,49 <curve_end> line,8,32 <curve_end> <loop_end> circle,21,31,31,21,41,31,31,41 <curve_end> <loop_end> <face_end> <sketch_end> add,31,63,31,31,31,1,0,0,0,1,0,0,0,1,53,31,31 <extrusion_end>

(14) circle,9,31,31,9,53,31,31,53 <curve_end> <loop_end> <face_end> <sketch_end> add,31,62,31,31,31,1,0,0,0,1,0,0,0,1,10,31,31 <extrusion_end>

(15) arc,9,15,9,9 <curve_end> line,15,9 <curve_end> arc,47,9,53,9 <curve_end> line,53,15 <curve_end> arc,53,47,53,53 <curve_end> line,47,53 <curve_end> arc,15,53,9,53 <curve_end> line,9,47 <curve_end> <loop_end> <face_end> <sketch_end> add,31,55,31,31,31,1,0,0,0,1,0,0,0,1,36,31,31 <extrusion_end>

(16) line,9,29 <curve_end> line,29,29 <curve_end> line,29,9 <curve_end> line,33,9 <curve_end> line,33,29 <curve_end> line,53,29 <curve_end> line,53,33 <curve_end> line,33,33 <curve_end> line,33,53 <curve_end> line,29,53 <curve_end> line,29,33 <curve_end> line,9,33 <curve_end> <loop_end> <face_end> <sketch_end> add,31,55,31,31,31,1,0,0,0,1,0,0,0,1,31,31,31 <extrusion_end>

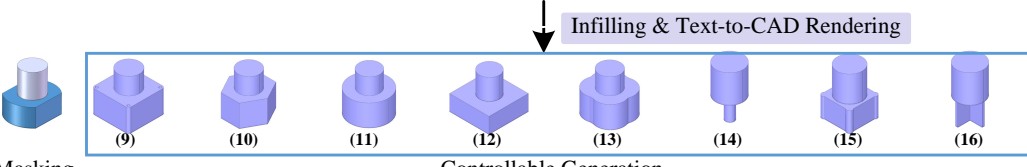

Masking        Infilling & Text-to-CAD Rendering

(9) (10) (11) (12) (13) (14) (15) (16)

Controllable Generation

Figure 11: Sketch-extrusion level generation. Given a CAD model, users can mask any of its sketch-extrusions for modifications. The masked sketch-extrusions are highlighted in blue.

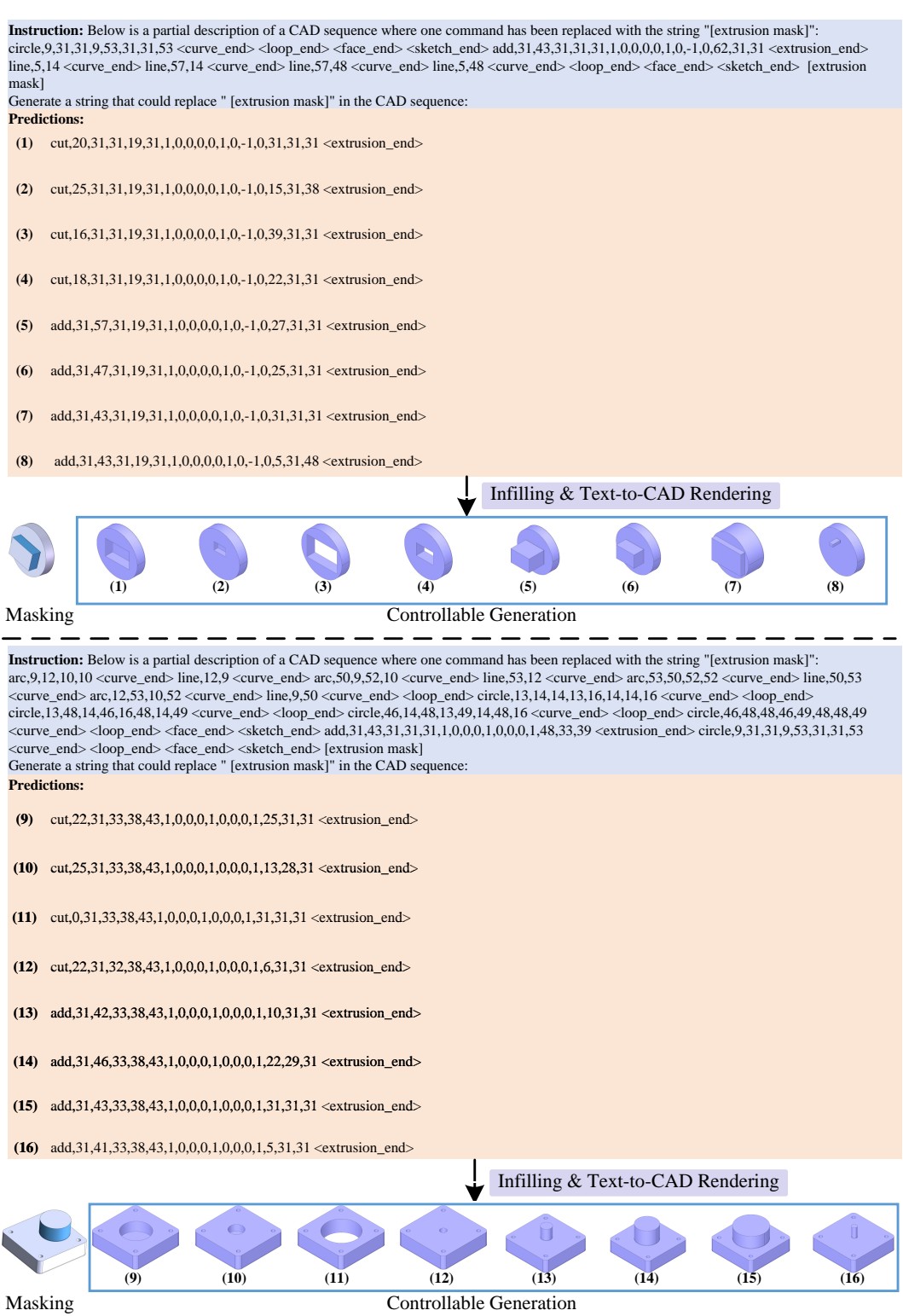

**Instruction:** Below is a partial description of a CAD sequence where one command has been replaced with the string "[extrusion mask]":
circle,9,31,31,9,53,31,31,53 <curve_end> <loop_end> <face_end> <sketch_end> add,31,43,31,31,31,1,0,0,0,0,1,0,-1,0,62,31,31 <extrusion_end>
line,5,14 <curve_end> line,57,14 <curve_end> line,57,48 <curve_end> line,5,48 <curve_end> <loop_end> <face_end> <sketch_end> [extrusion mask]
Generate a string that could replace " [extrusion mask]" in the CAD sequence:

**Predictions:**

(1)  cut,20,31,31,19,31,1,0,0,0,0,1,0,-1,0,31,31,31 <extrusion_end>

(2)  cut,25,31,31,19,31,1,0,0,0,0,1,0,-1,0,15,31,38 <extrusion_end>

(3)  cut,16,31,31,19,31,1,0,0,0,0,1,0,-1,0,39,31,31 <extrusion_end>

(4)  cut,18,31,31,19,31,1,0,0,0,0,1,0,-1,0,22,31,31 <extrusion_end>

(5)  add,31,57,31,19,31,1,0,0,0,0,1,0,-1,0,27,31,31 <extrusion_end>

(6)  add,31,47,31,19,31,1,0,0,0,0,1,0,-1,0,25,31,31 <extrusion_end>

(7)  add,31,43,31,19,31,1,0,0,0,0,1,0,-1,0,31,31,31 <extrusion_end>

(8)  add,31,43,31,19,31,1,0,0,0,0,1,0,-1,0,5,31,48 <extrusion_end>

Infilling & Text-to-CAD Rendering

(1) (2) (3) (4) (5) (6) (7) (8)

Masking                    Controllable Generation

**Instruction:** Below is a partial description of a CAD sequence where one command has been replaced with the string "[extrusion mask]":
arc,9,12,10,10 <curve_end> line,12,9 <curve_end> arc,50,9,52,10 <curve_end> line,53,12 <curve_end> arc,53,50,52,52 <curve_end> line,50,53
<curve_end> arc,12,53,10,52 <curve_end> line,9,50 <curve_end> <loop_end> circle,13,14,14,13,16,14,14,16 <curve_end> <loop_end>
circle,13,48,14,46,16,48,14,49 <curve_end> <loop_end> circle,46,14,48,13,49,14,48,16 <curve_end> <loop_end> circle,46,48,48,46,49,48,48,49
<curve_end> <loop_end> <face_end> <sketch_end> add,31,43,31,31,31,1,0,0,0,1,0,0,0,1,48,33,39 <extrusion_end> circle,9,31,31,9,53,31,31,53
<curve_end> <loop_end> <face_end> <sketch_end> [extrusion mask]
Generate a string that could replace " [extrusion mask]" in the CAD sequence:

**Predictions:**

(9)  cut,22,31,33,38,43,1,0,0,0,1,0,0,0,1,25,31,31 <extrusion_end>

(10)  cut,25,31,33,38,43,1,0,0,0,1,0,0,0,1,13,28,31 <extrusion_end>

(11)  cut,0,31,33,38,43,1,0,0,0,1,0,0,0,1,31,31,31 <extrusion_end>

(12)  cut,22,31,32,38,43,1,0,0,0,1,0,0,0,1,6,31,31 <extrusion_end>

(13)  add,31,42,33,38,43,1,0,0,0,1,0,0,0,1,10,31,31 <extrusion_end>

(14)  add,31,46,33,38,43,1,0,0,0,1,0,0,0,1,22,29,31 <extrusion_end>

(15)  add,31,43,33,38,43,1,0,0,0,1,0,0,0,1,31,31,31 <extrusion_end>

(16)  add,31,41,33,38,43,1,0,0,0,1,0,0,0,1,5,31,31 <extrusion_end>

Infilling & Text-to-CAD Rendering

(9) (10) (11) (12) (13) (14) (15) (16)

Masking                    Controllable Generation

Figure 12: Extrusion level generation. Given a CAD model, users can mask any of its extrusions for modifications. The masked extrusions are highlighted in blue.

**Instruction:** Below is a partial description of a CAD sequence where one command has been replaced with the string "[sketch mask]":
[sketch mask] add,31,63,31,31,31,1,0,0,0,1,0,0,0,1,50,31,31 <extrusion_end> circle,9,31,31,9,53,31,31,53 <curve_end> <loop_end> <face_end>
<sketch_end> add,31,50,31,31,63,1,0,0,0,1,0,0,0,1,12,31,31 <extrusion_end> line,10,19 <curve_end> line,31,7 <curve_end> line,52,19 <curve_end>
line,52,43 <curve_end> line,31,55 <curve_end> line,10,43 <curve_end> <loop_end> <face_end> <sketch_end>
cut,22,31,56,31,47,0,1,0,0,0,1,1,0,0,11,31,24 <extrusion_end>
Generate a string that could replace "[sketch mask]" in the CAD sequence:

**Predictions:**

(1) arc,9,11,9,9 <curve_end> line,11,9 <curve_end> arc,51,9,53,9 <curve_end> line,53,11 <curve_end> arc,53,51,53,53 <curve_end> line,51,53 <curve_end> arc,11,53,9,53 <curve_end> line,9,51 <curve_end> <loop_end> circle,11,12,12,11,14,12,12,14 <curve_end> <loop_end> circle,11,50,12,48,14,50,12,51 <curve_end> <loop_end> circle,48,12,50,11,51,12,50,14 <curve_end> <loop_end> circle,48,50,50,48,51,50,50,51 <curve_end> <loop_end> <face_end> <sketch_end>

(2) line,10,19 <curve_end> line,31,7 <curve_end> line,52,19 <curve_end> line,52,43 <curve_end> line,31,55 <curve_end> line,10,43 <curve_end> <loop_end> <face_end> <sketch_end>

(3) arc,9,20,10,10 <curve_end> line,20,9 <curve_end> arc,42,9,52,10 <curve_end> line,53,20 <curve_end> arc,53,42,52,52 <curve_end> line,42,53 <curve_end> arc,20,53,10,52 <curve_end> line,9,42 <curve_end> <loop_end> <face_end> <sketch_end>

(4) arc,9,24,10,21 <curve_end> line,13,20 <curve_end> arc,20,13,21,10 <curve_end> line,24,9 <curve_end> arc,38,9,41,10 <curve_end> line,42,13 <curve_end> arc,49,20,52,21 <curve_end> line,53,24 <curve_end> arc,53,38,52,41 <curve_end> line,49,42 <curve_end> arc,42,49,41,52 <curve_end> line,38,53 <curve_end> arc,24,53,21,52 <curve_end> line,20,49 <curve_end> arc,13,42,10,41 <curve_end> line,9,38 <curve_end> <loop_end> <face_end> <sketch_end>

Infilling & Text-to-CAD Rendering

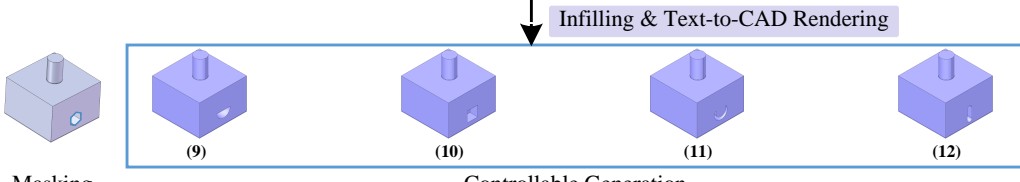

Masking        (1)    (2)    Controllable Generation    (3)    (4)

---

**Instruction:** Below is a partial description of a CAD sequence where one command has been replaced with the string "[sketch mask]":
line,9,9 <curve_end> line,9,53 <curve_end> line,53,53 <curve_end> line,53,9 <curve_end> <loop_end> <face_end> <sketch_end>
add,31,63,31,31,31,1,0,0,0,1,0,0,0,1,50,31,31 <extrusion_end> [sketch mask] add,31,50,31,31,63,1,0,0,0,1,0,0,0,1,12,31,31 <extrusion_end> line,10,19
<curve_end> line,31,7 <curve_end> line,52,19 <curve_end> line,52,43 <curve_end> line,31,55 <curve_end> line,10,43 <curve_end> <loop_end>
<face_end> <sketch_end> cut,22,31,56,31,47,0,1,0,0,0,1,1,0,0,11,31,24 <extrusion_end>
Generate a string that could replace " [sketch mask]" in the CAD sequence:

**Predictions:**

(5) arc,11,16,13,12 <curve_end> line,16,11 <curve_end> arc,46,11,49,12 <curve_end> line,51,16 <curve_end> arc,51,46,49,50 <curve_end> line,46,51 <curve_end> arc,16,51,13,50 <curve_end> line,11,46 <curve_end> <loop_end> arc,12,15,14,13 <curve_end> line,16,12 <curve_end> arc,46,12,48,13 <curve_end> line,50,15 <curve_end> arc,50,47,48,49 <curve_end> line,46,50 <curve_end> arc,16,50,14,49 <curve_end> line,12,47 <curve_end> <loop_end> <face_end> <sketch_end>

(6) arc,20,14,31,2 <curve_end> line,42,14 <curve_end> arc,42,48,31,60 <curve_end> line,20,48 <curve_end> <loop_end> <face_end> <sketch_end>

(7) line,10,19 <curve_end> line,31,7 <curve_end> line,52,19 <curve_end> line,52,43 <curve_end> line,31,55 <curve_end> line,10,43 <curve_end> <loop_end> line,11,19 <curve_end> line,31,8 <curve_end> line,51,19 <curve_end> line,51,43 <curve_end> line,31,54 <curve_end> line,11,43 <curve_end> <loop_end> <face_end> <sketch_end>

(8) arc,17,31,10,9 <curve_end> arc,25,13,31,10 <curve_end> arc,37,13,52,9 <curve_end> arc,45,31,46,33 <curve_end> arc,45,35,52,53 <curve_end> arc,37,49,31,52 <curve_end> arc,25,49,10,53 <curve_end> arc,17,35,16,33 <curve_end> <loop_end> <face_end> <sketch_end>

Infilling & Text-to-CAD Rendering

Masking      (5)    (6)    Controllable Generation    (7)    (8)

---

**Instruction:** Below is a partial description of a CAD sequence where one command has been replaced with the string "[sketch mask]":
line,9,9 <curve_end> line,9,53 <curve_end> line,53,53 <curve_end> line,53,9 <curve_end> <loop_end> <face_end> <sketch_end>
add,31,63,31,31,31,1,0,0,0,1,0,0,0,1,50,31,31 <extrusion_end> circle,9,31,31,9,53,31,31,53 <curve_end> <loop_end> <face_end> <sketch_end>
add,31,50,31,31,63,1,0,0,0,1,0,0,0,1,12,31,31 <extrusion_end> [sketch mask] cut,22,31,56,31,47,0,1,0,0,0,1,1,0,0,11,31,24 <extrusion_end>
Generate a string that could replace " [sketch mask]" in the CAD sequence:

**Predictions:**

(9) arc,3,45,31,17 <curve_end> line,59,45 <curve_end> <loop_end> <face_end> <sketch_end>

(10) line,9,9 <curve_end> line,53,9 <curve_end> line,53,53 <curve_end> line,9,53 <curve_end> <loop_end> <face_end> <sketch_end>

(11) arc,8,51,31,10 <curve_end> line,54,51 <curve_end> arc,50,51,31,21 <curve_end> line,12,51 <curve_end> <loop_end> <face_end> <sketch_end>

(12) arc,21,11,31,1 <curve_end> line,41,11 <curve_end> arc,41,51,31,61 <curve_end> line,21,51 <curve_end> <loop_end> <face_end> <sketch_end>

Infilling & Text-to-CAD Rendering

Masking      (9)    (10)    Controllable Generation    (11)    (12)

Figure 13: Sketch level generation. Given a CAD model, users can mask any of its sketches for modifications. The masked sketches are highlighted in blue.

**Instruction:** Below is a partial description of a CAD sequence where one command has been replaced with the string "[face mask] [face mask]":
line,11,7 <curve_end> line,51,7 <curve_end> line,51,55 <curve_end> line,11,55 <curve_end> <loop_end> <face_end> <sketch_end>
add,31,35,31,31,31,1,0,0,0,1,0,0,0,1,22,8,14 <extrusion_end> [face mask] [face mask] <sketch_end> add,31,41,10,16,35,1,0,0,0,1,0,0,0,1,9,30,33
<extrusion_end>
Generate a string that could replace "[face mask] [face mask]" in the CAD sequence:

**Predictions:**

(1)  line,0,26 <curve_end> line,16,26 <curve_end> line,16,36 <curve_end> line,0,36 <curve_end> <loop_end> <face_end> line,46,26 <curve_end>
      line,62,26 <curve_end> line,62,36 <curve_end> line,46,36 <curve_end> <loop_end> <face_end

(2)  circle,0,31,6,25,12,31,6,37 <curve_end> <loop_end> <face_end> circle,50,31,56,25,62,31,56,37 <curve_end> <loop_end> <face_end>

(3)  circle,23,5,29,0,34,5,29,11 <curve_end> <loop_end> <face_end> circle,23,57,29,51,34,57,29,62 <curve_end> <loop_end> <face_end>

(4)  line,17,3 <curve_end> line,45,3 <curve_end> line,45,17 <curve_end> line,17,17 <curve_end> <loop_end> <face_end> line,17,45 <curve_end>
      line,45,45 <curve_end> line,45,59 <curve_end> line,17,59 <curve_end> <loop_end> <face_end>

Infilling & Text-to-CAD Rendering

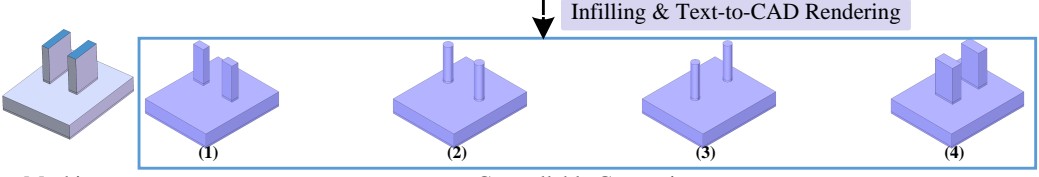

Masking          Controllable Generation

**Instruction:** Below is a partial description of a CAD sequence where one command has been replaced with the string "[face mask]":
line,11,7 <curve_end> line,51,7 <curve_end> line,51,55 <curve_end> line,11,55 <curve_end> <loop_end> <face_end> <sketch_end>
add,31,35,31,31,31,1,0,0,0,1,0,0,0,1,22,8,14 <extrusion_end> line,7,11 <curve_end> line,17,11 <curve_end> line,17,51 <curve_end> line,7,51
<curve_end> <loop_end> <face_end> [face mask] <sketch_end> add,31,41,10,16,35,1,0,0,0,1,0,0,0,1,9,30,33 <extrusion_end>
Generate a string that could replace " [face mask]" in the CAD sequence:

**Predictions:**

(5)  circle,35,31,45,21,55,31,45,41 <curve_end> <loop_end> circle,41,31,45,27,49,31,45,35 <curve_end> <loop_end> <face_end>

(6)  circle,42,31,51,22,60,31,51,40 <curve_end> <loop_end> <face_end>

(7)  arc,26,9,31,3 <curve_end> line,36,9 <curve_end> arc,36,53,31,59 <curve_end> line,26,53 <curve_end> <loop_end> <face_end>

(8)  arc,45,11,55,11 <curve_end> line,45,51 <curve_end> <loop_end> <face_end>

Infilling & Text-to-CAD Rendering

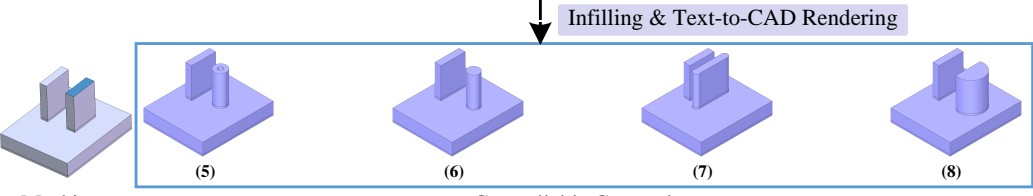

Masking          Controllable Generation

**Instruction:** Below is a partial description of a CAD sequence where one command has been replaced with the string "[face mask]":
line,11,7 <curve_end> line,51,7 <curve_end> line,51,55 <curve_end> line,11,55 <curve_end> <loop_end> <face_end> <sketch_end>
add,31,35,31,31,31,1,0,0,0,1,0,0,0,1,22,8,14 <extrusion_end> [face mask] line,43,13 <curve_end> line,55,13 <curve_end> line,55,51 <curve_end>
line,43,51 <curve_end> <loop_end> <face_end> <sketch_end> add,31,41,10,16,35,1,0,0,0,1,0,0,0,1,9,30,33 <extrusion_end>
Generate a string that could replace " [face mask]" in the CAD sequence:

**Predictions:**

(9)  line,7,11 <curve_end> line,16,11 <curve_end> line,16,27 <curve_end> line,7,27 <curve_end> <loop_end> <face_end>

(10)  line,7,11 <curve_end> line,35,11 <curve_end> line,35,51 <curve_end> line,7,51 <curve_end> <loop_end> <face_end>

(11)  circle,1,31,11,21,21,31,11,41 <curve_end> <loop_end> <face_end>

(12)  arc,13,7,15,5 <curve_end> line,17,7 <curve_end> line,17,31 <curve_end> arc,17,55,15,57 <curve_end> line,13,55 <curve_end> <loop_end>
      <face_end>

Infilling & Text-to-CAD Rendering

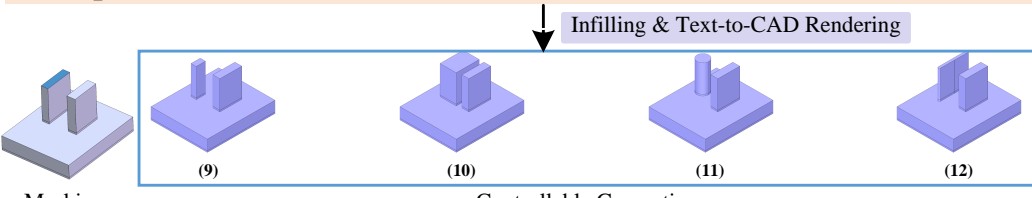

Masking          Controllable Generation

Figure 14: Face level generation. Given a sketch, users can mask one or multiple faces within it for modifications. The masked faces are highlighted in blue.

Figure 15: Loop level generation. Given a face, users can mask one or multiple loops within it for modifications. The masked loops are highlighted in blue.

**Instruction:** Below is a partial description of a CAD sequence where one command has been replaced with the string "[arc mask] [line mask] [arc mask] [line mask]":

line,18,2 <curve_end> line,44,2 <curve_end> line,44,60 <curve_end> line,18,60 <curve_end> <loop_end> [arc mask] [line mask] [arc mask] [line mask] <loop_end> <face_end> <sketch_end> add,31,34,31,31,31,1,0,0,0,0,1,0,-1,0,48,31,31 <extrusion_end>

Generate a string that could replace "[arc mask] [line mask] [arc mask] [line mask]" in the CAD sequence:

**Predictions:**

(1)  arc,29,17,31,11 <curve_end> line,33,17 <curve_end> arc,33,45,31,51 <curve_end> line,29,45 <curve_end>

(2)  arc,28,10,31,8 <curve_end> line,34,10 <curve_end> arc,34,52,31,54 <curve_end> line,28,52 <curve_end>

(3)  arc,24,18,31,15 <curve_end> line,38,18 <curve_end> arc,38,44,31,47 <curve_end> line,24,44 <curve_end>
(4)  arc,26,15,31,6 <curve_end> line,36,15 <curve_end> arc,36,47,31,52 <curve_end> line,26,47 <curve_end>

Infilling & Text-to-CAD Rendering

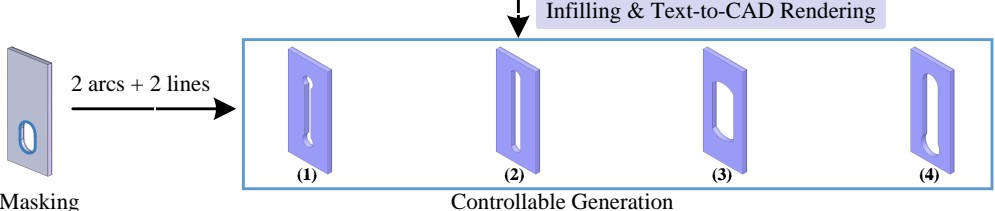

2 arcs + 2 lines

(1)    (2)    (3)    (4)

Masking          Controllable Generation

**Instruction:** Below is a partial description of a CAD sequence where one command has been replaced with the string "[arc mask] [line mask] [arc mask] [line mask] [arc mask] [line mask] [arc mask] [line mask]":

line,18,2 <curve_end> line,44,2 <curve_end> line,44,60 <curve_end> line,18,60 <curve_end> <loop_end> [arc mask] [line mask] [arc mask] [line mask] [arc mask] [line mask] [arc mask] [line mask] <loop_end> <face_end> <sketch_end> add,31,34,31,31,31,1,0,0,0,0,1,0,-1,0,48,31,31 <extrusion_end>

Generate a string that could replace "[arc mask] [line mask] [arc mask] [line mask] [arc mask] [line mask] [arc mask] [line mask]" in the CAD sequence:

**Predictions:**

(5)  arc,28,13,25,11 <curve_end> line,28,8 <curve_end> arc,34,8,37,11 <curve_end> line,34,13 <curve_end> arc,34,49,37,51 <curve_end> line,34,54 <curve_end> arc,28,54,25,51 <curve_end> line,28,49 <curve_end>

(6)  arc,27,43,23,31 <curve_end> line,27,19 <curve_end> arc,36,19,39,31 <curve_end> line,36,43 <curve_end> arc,36,50,36,53 <curve_end> line,33,55 <curve_end> arc,29,55,26,53 <curve_end> line,26,50 <curve_end>

(7)  arc,21,8,22,6 <curve_end> line,24,5 <curve_end> arc,38,5,40,6 <curve_end> line,41,8 <curve_end> arc,41,47,40,51 <curve_end> line,38,53 <curve_end> arc,24,53,22,51 <curve_end> line,21,47 <curve_end>

(8)  arc,26,11,23,9 <curve_end> line,26,6 <curve_end> arc,36,6,39,9 <curve_end> line,36,11 <curve_end> arc,36,26,39,28 <curve_end> line,36,31 <curve_end> arc,26,31,23,28 <curve_end> line,26,26 <curve_end>

Infilling & Text-to-CAD Rendering

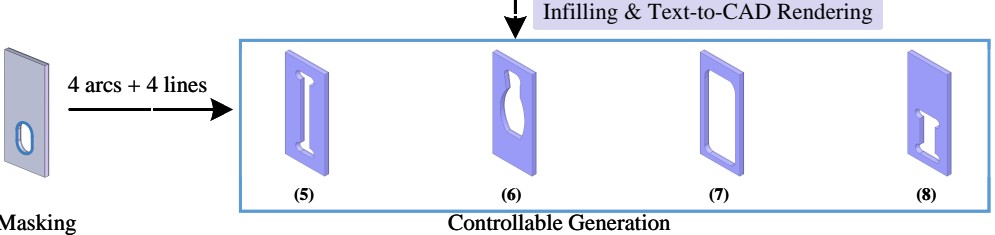

4 arcs + 4 lines

(5)    (6)    (7)    (8)

Masking          Controllable Generation

**Instruction:** Below is a partial description of a CAD sequence where one command has been replaced with the string "[line mask] [line mask] [line mask] [line mask]":

line,18,2 <curve_end> line,44,2 <curve_end> line,44,60 <curve_end> line,18,60 <curve_end> <loop_end> [line mask] [line mask] [line mask] [line mask] <loop_end> <face_end> <sketch_end> add,31,34,31,31,31,1,0,0,0,0,1,0,-1,0,48,31,31 <extrusion_end>

Generate a string that could replace "[line mask] [line mask] [line mask] [line mask]" in the CAD sequence:

**Predictions:**

(9)   line,23,7 <curve_end> line,39,7 <curve_end> line,39,55 <curve_end> line,23,55 <curve_end>

(10)  line,26,20 <curve_end> line,36,20 <curve_end> line,36,42 <curve_end> line,26,42 <curve_end>

(11)  line,24,7 <curve_end> line,38,7 <curve_end> line,38,31 <curve_end> line,24,31 <curve_end>

(12)  line,19,3 <curve_end> line,43,3 <curve_end> line,43,59 <curve_end> line,19,59 <curve_end>

Infilling & Text-to-CAD Rendering

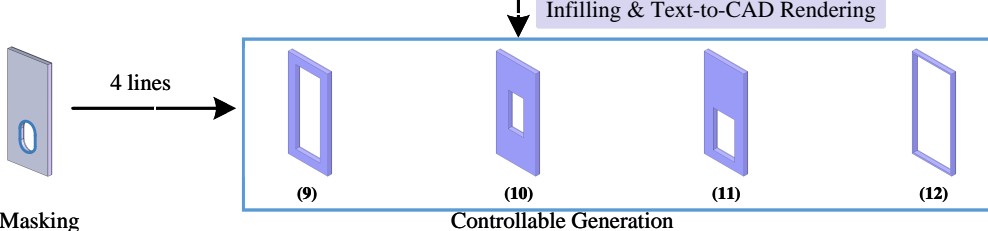

4 lines

(9)    (10)    (11)    (12)

Masking          Controllable Generation

Figure 16: Curve level generation. Given a loop, users can mask all curves within it and specify the curve type and number of newly generated loops. The masked curves are highlighted in blue.

**Instruction:** Below is a description of a CAD sequence:

**Predictions:**

**(1)** arc,9,26,14,14 <curve_end> arc,26,9,31,11 <curve_end> arc,36,9,48,14 <curve_end> arc,53,26,51,31 <curve_end> arc,53,36,48,48 <curve_end> arc,36,53,31,51 <curve_end> arc,26,53,14,48 <curve_end> arc,9,36,11,31 <curve_end> <loop_end> circle,11,25,16,20,22,25,16,31 <curve_end> <loop_end> circle,11,37,16,31,22,37,16,42 <curve_end> <loop_end> circle,18,16,23,11,28,16,23,21 <curve_end> <loop_end> circle,18,46,23,41,28,46,23,51 <curve_end> <loop_end> circle,27,31,31,27,35,31,31,35 <curve_end> <loop_end> circle,34,16,39,11,44,16,39,21 <curve_end> <loop_end> circle,34,46,39,41,44,46,39,51 <curve_end> <loop_end> circle,40,25,46,20,51,25,46,31 <curve_end> <loop_end> circle,40,37,46,31,51,37,46,42 <curve_end> <loop_end> <face_end> <sketch_end> add,31,37,31,31,31,1,0,0,0,0,1,0,-1,0,62,31,31 <extrusion_end>

**(2)** line,5,13 <curve_end> line,57,13 <curve_end> line,57,49 <curve_end> line,5,49 <curve_end> <loop_end> circle,23,31,31,23,39,31,31,39 <curve_end> <loop_end> <face_end> <sketch_end> add,31,63,31,31,31,1,0,0,0,1,0,0,0,1,23,31,31 <extrusion_end> circle,9,31,31,9,53,31,31,53 <curve_end> <loop_end> circle,20,31,31,20,42,31,31,42 <curve_end> <loop_end> <face_end> <sketch_end> add,31,43,31,31,63,1,0,0,0,1,0,0,0,1,16,31,31 <extrusion_end>

**(3)** arc,13,42,2,31 <curve_end> line,13,20 <curve_end> arc,21,21,25,23 <curve_end> arc,29,21,31,20 <curve_end> arc,33,21,37,23 <curve_end> line,41,21 <curve_end> arc,49,20,60,31 <curve_end> line,49,42 <curve_end> arc,41,41,37,39 <curve_end> arc,33,41,31,42 <curve_end> arc,29,41,25,39 <curve_end> line,21,41 <curve_end> <loop_end> circle,5,31,10,27,14,31,10,35 <curve_end> <loop_end> circle,48,31,52,27,57,31,52,35 <curve_end> <loop_end> <face_end> <sketch_end> add,31,35,31,31,31,1,0,0,0,0,1,0,-1,0,47,31,31 <extrusion_end>

**(4)** line,9,9 <curve_end> line,16,9 <curve_end> line,16,13 <curve_end> line,46,13 <curve_end> line,46,9 <curve_end> line,53,9 <curve_end> line,53,16 <curve_end> line,49,16 <curve_end> line,49,46 <curve_end> line,53,46 <curve_end> line,53,53 <curve_end> line,46,53 <curve_end> line,46,49 <curve_end> line,16,49 <curve_end> line,16,53 <curve_end> line,9,53 <curve_end> line,9,46 <curve_end> line,13,46 <curve_end> line,13,16 <curve_end> line,9,16 <curve_end> <loop_end> <face_end> <sketch_end> add,31,39,31,31,31,1,0,0,0,0,1,0,-1,0,31,48,14 <extrusion_end>

**(5)** arc,7,28,31,11 <curve_end> line,55,28 <curve_end> arc,51,51,31,44 <curve_end> line,11,51 <curve_end> <loop_end> circle,26,28,31,22,36,28,31,33 <curve_end> <loop_end> <face_end> <sketch_end> add,31,47,31,31,31,1,0,0,0,0,1,0,-1,0,50,31,41 <extrusion_end>

**(6)** line,9,9 <curve_end> line,53,9 <curve_end> line,53,53 <curve_end> line,9,53 <curve_end> <loop_end> <face_end> <sketch_end> add,31,39,31,31,31,1,0,0,0,1,0,0,0,1,62,31,31 <extrusion_end> line,11,7 <curve_end> line,51,7 <curve_end> line,51,55 <curve_end> line,51,7 <curve_end> <loop_end> <face_end> <sketch_end> add,31,37,31,31,39,1,0,0,0,1,0,0,0,1,42,31,31 <extrusion_end>

**(7)** arc,20,9,31,2 <curve_end> line,42,9 <curve_end> arc,42,11,42,13 <curve_end> line,40,14 <curve_end> arc,36,15,35,16 <curve_end> line,35,17 <curve_end> arc,35,21,34,23 <curve_end> line,35,24 <curve_end> line,35,49 <curve_end> arc,35,56,31,60 <curve_end> line,27,56 <curve_end> line,27,49 <curve_end> arc,27,23,28,22 <curve_end> line,27,21 <curve_end> arc,27,17,27,16 <curve_end> line,26,15 <curve_end> arc,22,14,20,13 <curve_end> line,20,11 <curve_end> <loop_end> circle,28,8,31,6,34,8,31,11 <curve_end> <loop_end> circle,29,52,31,50,33,52,31,55 <curve_end> <loop_end> <face_end> <sketch_end> add,31,32,31,31,31,1,0,0,0,1,0,0,0,1,28,26,45 <extrusion_end>

**(8)** circle,9,31,31,9,53,31,31,53 <curve_end> <loop_end> <face_end> <sketch_end> add,31,57,31,31,31,1,0,0,0,0,1,0,-1,0,23,31,31 <extrusion_end> line,8,30 <curve_end> line,25,10 <curve_end> line,50,12 <curve_end> line,54,32 <curve_end> line,37,52 <curve_end> line,12,50 <curve_end> <loop_end> <face_end> <sketch_end> add,31,41,31,31,31,1,0,0,0,0,1,0,-1,0,37,31,31 <extrusion_end>

**(9)** line,2,20 <curve_end> line,2,42 <curve_end> line,31,42 <curve_end> line,60,42 <curve_end> line,60,20 <curve_end> line,31,20 <curve_end> <loop_end> circle,9,31,15,25,21,31,15,37 <curve_end> <loop_end> circle,41,31,47,25,53,31,47,37 <curve_end> <loop_end> <face_end> <sketch_end> add,31,46,31,31,31,1,0,0,0,0,1,0,-1,0,47,31,31 <extrusion_end>

**(10)** arc,0,31,3,28 <curve_end> line,6,31 <curve_end> arc,56,31,59,28 <curve_end> line,62,31 <curve_end> line,62,34 <curve_end> line,0,34 <curve_end> <loop_end> <face_end> <sketch_end> add,31,36,31,31,31,1,0,0,0,1,0,0,0,1,22,14,34 <extrusion_end>

**(11)** line,4,14 <curve_end> line,58,14 <curve_end> line,58,48 <curve_end> line,4,48 <curve_end> <loop_end> line,19,21 <curve_end> line,22,21 <curve_end> line,22,26 <curve_end> line,19,26 <curve_end> <loop_end> line,21,36 <curve_end> line,25,36 <curve_end> line,25,40 <curve_end> line,21,40 <curve_end> <loop_end> circle,43,31,48,26,53,31,48,36 <curve_end> <loop_end> <face_end> <sketch_end> add,31,43,31,31,31,1,0,0,0,1,0,0,0,1,26,14,42 <extrusion_end>

**(12)** line,7,11 <curve_end> line,31,11 <curve_end> line,55,11 <curve_end> line,55,27 <curve_end> line,36,27 <curve_end> line,36,35 <curve_end> line,55,35 <curve_end> line,55,51 <curve_end> line,31,51 <curve_end> line,7,51 <curve_end> line,7,35 <curve_end> line,21,35 <curve_end> line,21,27 <curve_end> line,7,27 <curve_end> <loop_end> <face_end> <sketch_end> add,31,45,31,31,31,1,0,0,0,0,1,0,-1,0,28,14,45 <extrusion_end>

↓ Text-to-CAD Rendering

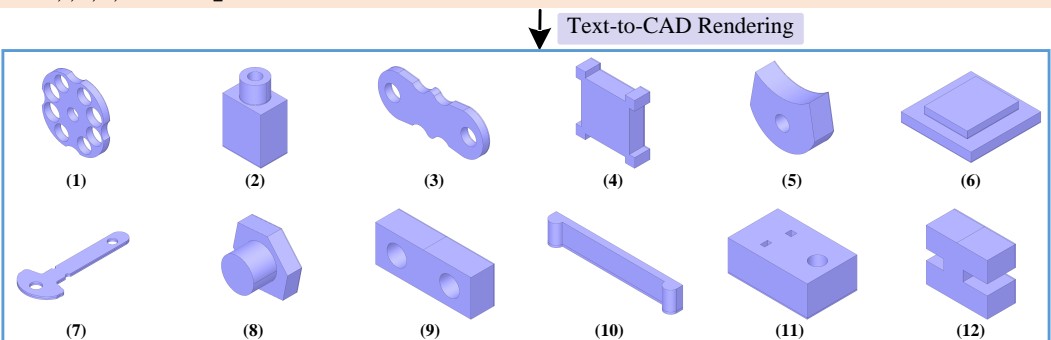

(1)    (2)    (3)    (4)    (5)    (6)

(7)    (8)    (9)    (10)    (11)    (12)

Figure 17: Unconditional CAD generation. Here, users can generate CAD models without any conditional limitations.

