# OpenReview forum: "FlexCAD: Unified and Versatile Controllable CAD Generation with Fine-tuned Large Language Models"
_ICLR.cc/2025/Conference — ICLR 2025 Poster_

### Official Review · Reviewer_4Ahh · 2024-10-19

**Soundness:** 3
**Presentation:** 3
**Contribution:** 3
**Rating:** 6
**Confidence:** 3

**Summary:**

This paper presents FlexCAD, a unified model for controllable CAD generation, enhancing the ability to generate CAD models based on user intent. Unlike existing methods requiring separate models for different controls, FlexCAD enables controllable generation across multiple CAD construction hierarchies (e.g., sketch, extrusion, face). The authors fine-tune large language models (LLMs) and represent CAD models as structured text, abstracting each hierarchy into a sequence of text tokens for better comprehension. A hierarchy-aware masking strategy is introduced, masking specific fields during training to predict the masked parts based on user input during inference. In general, this paper is well organized.

**Strengths:**

+ Novel use of LLMs for CAD design
+ The authors propose a unified framework based on fine-tuning LLMs to bridge LLMs and CAD models, including dividing CAD models into multiple hierarchies and representing CAD models with sequential tokens and
+ The authors introduce a hierarchy-aware masking strategy to achieve controllable generation based on the mask-and-prediction method
+ Experimental results show the proposed method achieves good results and verify the effectiveness of each component

**Weaknesses:**

- The motivation of using LLMs for CAD generation is not very clear. The authors are recommended to clarify why LLMs are the good choice to do this task.

- The ablation study in Tab 3 is clear to make a comparison among different foundation models, pre-training, and fine-tuning. However, could LLaMA-3 do CAD generation in a zero-shot manner? How helpful is the proposed fine-tuning method compared with the zero-shot setting?

- The evaluation for prompt-result alignment is unclear, e.g., intent understanding and instruction following.

- Some experimental details may be missing, e.g., the instructions used to prompt GPT-4o to generate CAD models.

**Questions:**

- Does the proposed method inherit some advantages of LLMs, such as instruction following and in-context learning?

- The proposed method assumes users skilled at the defined hierarchies so that they can transform their intent into mask-based instructions during inference. Can users express their intent using natural language and then the transformation task is given to LLMs?

- Does the fine-tuned model strictly follow the grammar of CAD structured text? If not, is there any failure case which may not be parsed into a valid CAD model? How about the percentage?

- Compared with prior work, is there any drawback to using LLMs? How about the efficiency?

---

> ### Author Response · Authors · 2024-11-19
> **Response to Reviewer 4Ahh (1/2)**
>
> **Q1: Why are LLMs the good choice to do this task?**
> A1: As mentioned in Lines 68–76, the main reasons are as follows:
> (1) LLMs have demonstrated remarkable success in handling diverse user queries with a single, unified model.
> Thus, we explored and successfully introduced LLM-based FlexCAD to achieve controllable generation across seven different hierarchies within a unified framework.
> (2) LLMs may have acquired CAD-related knowledge during pre-training by learning CAD-specific code, such as JSCAD.
> As shown in Table 2, this pre-training knowledge contributes to performance gains.
> (3) Before the advent of LLMs, small transformer-based models were explored for CAD-related generation tasks, highlighting the potential of transformers for these tasks from a different perspective.
> (4) Leveraging pre-trained weights, our FlexCAD inherits instruction-following capabilities of LLMs, allowing it to understand instructions that never appeared during fine-tuning (see A5 for more details).
>
> **Q2: Could LLaMA-3 do CAD generation in a zero-shot manner? How helpful is the proposed fine-tuning method compared with the zero-shot setting?**
> A2: LLaMA-3 (both 7B and 72B) cannot do CAD generation in a zero-shot manner.
> Without fine-tuning, the output format is largely incompatible with our CAD textual representation.
> For instance, point coordinates for circles, arcs and lines are often incomplete, and the end of each hierarchy is always missing.
> In contrast, with our proposed fine-tuning method, the output fully adheres to the expected format, and the generated CAD text can be successfully rendered, demonstrating the effectiveness of fine-tuning.
>
> **Q3: The evaluation for prompt-result alignment is unclear, e.g., intent understanding and instruction following.**
> A3:  (1) The prompt-result alignment can be evaluated by checking whether the type and number of hierarchies in the generated CAD match those specified in the prompt.  We utilize string matching for automated evaluation.
> The overall score across all hierarchies is over 98%, with the score for the most challenging level, the curve level, reaching 95%.
> We did not include this in the original version as it is relatively simple for LLMs, while we will add it in the revision for comprehensive evaluation.
> (2) We also use the Realism metric to evaluate whether the newly generated parts under the corresponding prompts integrate seamlessly with the remaining unmasked elements without conflict.
> As shown in Table 1, our FlexCAD significantly surpasses baselines in terms of the Realism metric.
>
> **Q4: The instructions used to prompt GPT-4o.**
> Thank you for pointing this out. We have added a detailed GPT-4o prompt example in Fig. 18 in our paper.
>
> **Q5: Does the proposed method inherit some advantages of LLMs, such as instruction following and in-context learning?**
> A5: Yes, it does.
> With pre-trained knowledge, our FlexCAD inherits the instruction-following and in-context learning capabilities of LLMs,
> allowing it to understand instructions that never appeared during fine-tuning.
> To provide further evidence, we perform additional experiments.
> Taking sketch-level generation as an example,
> we add additional instructions (e.g., *'Ensure that the sketch contains at least one circle'*) at the end of the original instructions.
> Consequently, the proportion of newly generated sketches containing circles surges from 37% to 89% out of a total of 1000 sketches generated.
> Moreover, for in-context learning, when we provide FlexCAD with five exemplars where all the answer sketches contain circles,
> the proportion of newly generated sketches containing circles increases from 37% to 65%.
>
> **Q6: Can users express their intent using natural language and then the transformation task is given to LLMs?**
> Currently, users can primarily express their intent for controllable generation by masking the corresponding field.
> Users can also utilize natural language (i.e., text) to instruct our FlexCAD to meet their intent, as discussed in A5.
> However, achieving complex text-to-CAD alignment remains challenging.
> Specifically, there is a lack of relevant textual descriptions for each CAD model,
> such as 'a regular hexagon' or 'a desk with four legs', which are necessary for fine-tuning.
> We believe that by constructing a comprehensive text-to-CAD dataset and further fine-tuning our FlexCAD in the future, users will be able to more easily and effectively guide CAD generation using natural language.

---

> > ### Author Response · Authors · 2024-11-19
> > **Response to Reviewer 4Ahh (2/2)**
> >
> > **Q7: Does the fine-tuned model strictly follow the grammar of CAD structured text?  Any failure case? How about the percentage?**
> > Yes, the fine-tuned model strictly follows the grammar of CAD structured text, including curve representations (i.e., lines, arcs, and circles), the end of each hierarchy, and extrusion operations.
> > (Please refer to Fig. 10–Fig. 17 for further validation.)
> > However, even when the grammar is entirely correct, the numerical values in curves might not form a valid sketch.
> > For instance, if two points overlap, they cannot form a valid line.
> > Similarly, invalid extrusion values may prevent the sketch from being successfully transformed into a 3D CAD model.
> > As mentioned in Line 299, we use Prediction Validity (PV) to denote the overall prediction validity, ensuring CAD texts can be rendered into 3D shapes rather than just 2D sketches or nothing.
> > As shown in Table 1, for sketch-level and extrusion-level controllable generation, the percentage of cases that cannot be parsed into a valid CAD model is approximately **6.6%–6.7%**, significantly lower than baselines.
> >
> > **Q8: Compared with prior work, is there any drawback to using LLMs? How about the efficiency?**
> > We agree with you that the efficiency is an important issue.
> > Actually, we have compared the inference time between our method and baselines, as mentioned in Lines 778-786.
> > Compared to baselines, i.e., SkexGen and Hnc-cad, both of which use multiple transformers for inference,
> > our FlexCAD conducts inference within a single framework.
> > The inference time of our FlexCAD (based on Llama-3-8B) is slightly higher than that of baselines (see Lines 778-786).
> > While the task of controllable CAD generation is not particularly demanding in terms of real-time inference requirements,
> > we believe the slight increase in inference time for our FlexCAD (based on Llama-3-8B) is acceptable, given its promising performance.
> > Besides, the parallelization and quantization techniques can be further applied to shorten inference time in the future.

---

> > > ### Comment · Reviewer_4Ahh · 2024-12-02
> > >
> > > Thanks for the work and thoughtful rebuttal.
> > >
> > > I appreciate the authors' efforts in addressing my concerns during the rebuttal process. Given the contributions of this work, I am inclined to maintain my positive score.

---

### Official Review · Reviewer_DaWS · 2024-11-02

**Soundness:** 3
**Presentation:** 3
**Contribution:** 2
**Rating:** 6
**Confidence:** 3

**Summary:**

This paper proposes a unified CAD generation model that leverages large language models with a hierarchy-aware masking strategy, achieving fine-grained controllability across multiple CAD construction hierarchies and outperforming existing methods in generation quality and user-intent alignment.
In detail, FlexCAD represents CAD models as structured text, transforming CAD construction hierarchies into sequences of text tokens, which improves comprehension by LLMs. During training, the hierarchy-aware masking replaces specific fields in CAD text with a mask token, enabling the model to predict missing parts based on user intent.
The model demonstrates significant performance improvements over existing methods like SkexGen and Hnc-CAD.

**Strengths:**

1. The proposed FlexCAD model effectively enables controllable CAD generation across all hierarchies, addressing a significant gap in CAD modeling by unifying control within a single framework. It achieves superior performance on the DeepCAD dataset compared to existing baselines, highlighting the method's robustness and effectiveness.
2. The paper is well-organized and clearly written, making the methodology and contributions easy to understand and follow.
3. The authors provide extensive ablation studies to analyze the impact of LLM pre-training and hierarchy-aware tokenization. These experiments demonstrate that fine-tuning pre-trained LLMs with LoRA yields the best results, underscoring the advantages of leveraging pre-trained models for CAD representation—a promising insight for future research in CAD and design automation.

**Weaknesses:**

1. The evaluation is limited to the DeepCAD dataset, which restricts the generalizability of the results. Testing FlexCAD on additional CAD datasets would better demonstrate its robustness and adaptability across diverse design scenarios.
2. The paper lacks references to relevant recent work, such as “OpenECAD: An Efficient Visual Language Model for Editable 3D-CAD Design” and “CadVLM: Bridging Language and Vision in the Generation of Parametric CAD Sketches,” which address related challenges in CAD generation and infilling. Including these would strengthen the context and positioning of the work.

**Questions:**

1. During training, what is the average mask ratio per training sample, and is there any observed relationship between performance and the mask ratio?
2. Given that a single masked input can produce multiple possible outputs, as shown in Figure 5, which output is selected for the final evaluation?

---

> ### Author Response · Authors · 2024-11-19
> **Response to Reviewer DaWS**
>
> **Q1: Additional CAD datasets.**
> A1: We appreciate your feedback and will evaluate additional CAD datasets once a suitable one is found or collected.
> Currently, DeepCAD is the only dataset suitable for evaluation, and the detailed reasons are explained below:
> (1) DeepCAD is a large-scale 3D CAD dataset, consisting of more than 178k samples.
> (2) Compared to 2D sketch datasets, DeepCAD better reflects our controllability, not only on sketches but also on extrusions.
> (3) Compared to other 3D CAD datasets, such as the ABC dataset (Abc: A Big CAD Model Dataset for Geometric Deep Learning, CVPR 2019),
> DeepCAD includes sketch-and-extrusion sequences, which align with the CAD design process in commercial CAD tools like SolidWorks and AutoCAD.
> (4) Given the characteristics of DeepCAD, baselines (i.e., SkexGen [ICML 2022] and Hnc-cad [ICML 2023]) also use **only DeepCAD**,
> making it more convenient to compare performance.
> (5) Beyond our controllable generation task, for other tasks such as Text-to-CAD,
> the paper (Generating Sequential CAD Models from Beginner-to-Expert Level Text Prompts, NeurIPS 2024) also uses **only DeepCAD**.
>
> **Q2: Including OpenECAD and CadVLM would strengthen the context and positioning of the work.**
> A2: Good suggestion. We will cite and discuss them in the revision.
> Notably, unlike our focus on local controllable generation, OpenECAD and CadVLM focus on the image-to-CAD task using VLMs.
>
> **Q3: During training, what is the average mask ratio per training sample, and is there any observed relationship between performance and the mask ratio?**
> A3: Instead of masking a fixed ratio for each sample, our masking mechanism is to mask a hierarchy from CAD, sketch-extrusion, extrusion, sketch, face, loop and curve, as shown in Fig. 4.
> The average mask ratios per training sample for the CAD, sketch-extrusion, extrusion, sketch, face, loop and curve hierarchies are approximately 100%, 49.8%, 7.5%, 42.2%, 40.6%, 9.79% and 8.42%, respectively.
> Generally, as reported in Table 5, coarser-grained hierarchies (e.g., CAD and sketch-extrusion) with larger mask ratios outperform finer-grained hierarchies (e.g., face, loop, and curve) with smaller mask ratios.
> This is likely because the new generations produced by our FlexCAD for the masked part strictly align with the remaining (unmasked) conditions.
> In other words, the new generations avoid potential conflict with the remaining conditions in 3D space.
> As the mask ratio decreases, the remaining conditions (i.e., constraints for the masked part) become longer and more complex, thereby limiting the overall quality and diversity of the masked part.
>
> **Q4: As shown in Figure 5, which output is selected for the final evaluation?**
> A4: For the quantitative comparison (i.e., Table 1), as mentioned in Lines 315-316, we randomly selected 1,000 CAD models from the test set. For each method, we generated 10 predictions per CAD model,
> resulting in a total of 10,000 CAD models, **all of which were used for the final evaluation**.
> For the qualitative comparison (i.e., Fig. 5), we also generated 10 predictions for each CAD model and randomly selected four newly predicted models for each method, as mentioned in Line 365.

---

### Official Review · Reviewer_zSTF · 2024-11-03

**Soundness:** 3
**Presentation:** 3
**Contribution:** 2
**Rating:** 3
**Confidence:** 2

**Summary:**

This paper proposed, flexCAD, a unified and versatile model for controllable CAD generation across all hierarchies using LLMs. FlexCAD addresses various limitations in existing CAD generation models. Previous models typically lack comprehensive control across CAD hierarchies and require separate models for different tasks. The proposed model leverages a structured text representation of CAD models and a hierarchy-aware masking strategy to control the generation across various levels (e.g., sketch, face, loop, extrusion). Experiments show that FlexCAD outperforms existing models on various metrics.

Contributions:
- FlexCAD is the first to leverage LLMs for controllable CAD generation.
- FlexCAD greatly improves generation quality and controllability, showing its effectiveness on the sketch-level and extrusion-level controllable generation tasks.

**Strengths:**

- FlexCAD improves generation quality and controllability in some metrics and demonstrates its effectiveness on the sketch-level and extrusion-level controllable generation tasks.

**Weaknesses:**

- Novelty is limited. The training methods being used are very standard tuning methods for LLM.
- FlexCAD is the first to leverage LLMs for controllable CAD generation (based on the author's statement). Previous baseline "SkexGen: Autoregressive Generation of CAD Construction Sequences with Disentangled Codebooks" and "Hierarchical Neural Coding for Controllable CAD Model Generation (ICML 2023)" already use transformer architecture.
- It would be stronger if FlexCAD can show improvement with larger margin given FlexCAD is not the best among all metrics.
- No discussion on ethics and potential risk of this research.

**Questions:**

- Can the authors provide some analysis on why full model fine-tuning performs worse than LoRA on some metrics?
- For FlexCAD, does it use few-shot or zero-shot at inference time?

---

> ### Author Response · Authors · 2024-11-19
> **Response to Reviewer zSTF (1/2)**
>
> **Q1: Novelty is limited. The training methods are very standard for tuning LLM.**
> A1: Actually, this paper is a novel use of LLMs for CAD design, aligning with the feedback from reviewer 4Ahh.
> Here, we highlight our novelty and explain the training methods as follows:
> (1) Baselines (SkexGen and Hnc-cad) generally utilize VQ-VAE in combination with small transformers to produce numerical codes for CAD generation.
> On the other hand, we shift this generation paradigm by introducing LLMs for controllable CAD generation.
> To further clarify our novelty, we will explain the difference between our LLM-based FlexCAD and transformers of baselines in A2.
> (2) Yes, the training methods are very standard (see Line 261).
> We deliberately maintain standard training methods with minimal changes,
> considering this a key advantage of our approach. The reason is threefold:
> > - **Demonstrating the Power of Simplicity.**
> Our simple method significantly outperforms baselines
> that rely on complex models and training techniques. By showcasing this to the community,
> we aim to encourage future work to build on this foundation,
> advancing the field with practical and effective solutions.
> > - **Evidence from Proven Approaches.**
> In other domains, similar efforts using standard LLM training methods [Refs A, B, C, D, E] have shown significant impact.
> We believe our approach will similarly transform the CAD generation domain.
> Ref A: PointLLM: Empowering Large Language Models to Understand Point Clouds (ECCV2024)
> Ref B: Motiongpt: Finetuned LLMs Are General-Purpose Motion Generators (AAAI2024)
> Ref C: AutoTimes: Autoregressive Time Series Forecasters via Large Language Models (Neurips2024)
> Ref D: UniAudio: Towards Universal Audio Generation with Large Language Models (ICML2024)
> Ref E: Fine-Tuned Language Models Generate Stable Inorganic Materials as Text (ICLR2024)
> > - **Alignment with AGI Goals.** Achieving artificial general intelligence (AGI) requires models capable of generalizing knowledge across diverse tasks and domains.
> The autoregressive Transformer and the next token prediction employed by LLMs are considered as promising methods towards achieving AGI.
> By using standard training methods of LLMs with minimal modifications, our work in CAD design can seamlessly integrate with efforts in other domains, contributing to the broader pursuit of AGI.
>
>
> **Q2:  LLMs vs. Transformers in SkexGen and Hnc-cad**
> A2:  While both our method and baselines utilize transformers, there are significant distinctions.
> Our approach primarily involves fine-tuning LLMs, whereas baselines rely on VQ-VAE frameworks.
> The differences are introduced below:
> (1) As mentioned in Line 94, unlike SkexGen and Hnc-cad, which require multi-stage training to train multiple transformers,
> our FlexCAD has only one transformer decoder and achieves end-to-end training in a unified framework. (Specifically, Skexgen has five transformers: topology encoder, geometry encoder, extrusion encoder, sketch decoder and extrusion decoder.
> Hnc-cad has three transformers: model encoder, code tree generator and model generator.)
> (2) As mentioned in Lines 60-65, unlike SkexGen and Hnc-cad, which have limited controllable abilities, our LLM-based FlexCAD achieves controllable CAD generation across all hierarchies,
> including sketch-extrusion, extrusion, sketch, face, loop, and curve. Besides, as shown in Table 1 and Fig. 5, compared to SkexGen and Hnc-cad, our FlexCAD enhances both generation quality and controllability.
> (3) The parameter sizes are not on the same level; the parameter count in the largest transformer used in SkexGen and Hnc-cad is no more than 10M, while that of our smallest LLM is more than 8B.
> These parameters of LLMs are used to retain pre-trained knowledge, which has been demonstrated to enhance performance (see Table 2).
> (4)  Unlike SkexGen and Hnc-cad, whose input and output are in numerical format,
> we transform the CAD representation into a textual format that is easily understood by both LLMs and users.
> (5) Relying on pre-trained knowledge, our FlexCAD inherits the instruction-following and in-context learning abilities of LLMs, enabling it to comprehend instructions that never appear during fine-tuning,
> a capability that small transformers in SkexGen and Hnc-cad cannot achieve.
> To provide further evidence, we conduct additional experiments.
> Taking sketch-level generation as an example,
> we add additional instructions (e.g., *'Ensure that the sketch contains at least one circle'*) at the end of the original instructions.
> As a result, the proportion of newly generated sketches containing circles increases from 37% to 89% out of a total of 1000 sketches generated.
> Furthermore, for in-context learning, when we provide FlexCAD with five exemplars where all the answer sketches contain circles,
> the proportion of newly generated sketches containing circles increases from 37% to 65%.

---

> > ### Author Response · Authors · 2024-11-19
> > **Response to Reviewer zSTF (2/2)**
> >
> > **Q3:  Not the best among all metrics**
> > A3: SkexGen (ICML 2022) outperforms our FlexCAD on part of the Unique and Novel metrics.
> > Similarly, as noted in the Hnc-cad paper (ICML 2023), SkexGen also outperforms Hnc-cad on these metrics.
> > The Hnc-cad paper attributes this to **SkexGen's inability to generate very complex CAD models**,
> > which aligns with our experimental observations (As shown in Fig. 5, when generating complex CAD models, SkexGen tends to produce unrealistic ones, which are still counted as novel and unique during evaluation).
> > On the other hand, our FlexCAD achieves the best performance across the other five metrics, further demonstrating its effectiveness.
> >
> > **Q4:  Ethics and potential risk**
> > A4: Thanks for pointing out this.
> > The data used in this work is tailored for creating and modifying CAD models. Due to its specialized
> > nature, the misuse risk is naturally minimized, ensuring that the developed methods primarily benefit
> > design and engineering tasks. In this work, we have invited crowd workers to evaluate the quality
> > of generated CAD models. We conducted this work in accordance with ethical guidelines to ensure
> > that participants were treated fairly, respectfully, and safely throughout the process. We took steps
> > to protect the privacy of crowd workers by not collecting personally identifiable information. The
> > data annotated by the crowd workers was used only for research purpose related to improving CAD
> > generating techniques.
> >
> > **Q5:  Why does full model fine-tuning perform worse than LoRA on some metrics?**
> > A5: (1) We load pre-trained weights when fine-tuning LLMs using LoRA. Since the pre-trained weights may contain useful CAD-related knowledge,
> > this knowledge might be forgotten as the number of training epochs increases when training the full model.
> > In contrast, by using LoRA, we can better **retain the pre-trained knowledge by freezing most of the parameters**.
> > (2) As demonstrated by CrystalLLM (Fine-tuned language models generate stable inorganic materials as text, ICLR2024), comparing training full model, **using LoRA can help reduce overfit**.
> > (3) It is also common in the original LoRA paper and other works that full model fine-tuning performs worse than LoRA on some metrics, such as:
> > Make Pre-trained Model Reversible: From Parameter to Memory Efficient Fine-Tuning (NeurIPS 2023)
> > Tracking Meets LoRA: Faster Training, Larger Model, Stronger Performance (ECCV2024).
> >
> >
> > **Q6:  Using few-shot or zero-shot at inference time?**
> > A6: Zero-shot. As shown in Figs. 10–17, we present prompt examples across all hierarchies during inference, all of which are zero-shot.

---

> > ### Comment · Reviewer_zSTF · 2024-11-24
> >
> > Thank you for the detailed clarification. I acknowledge that fine-tuning LLMs is a new approach within this specific domain and that it has led to improvements in various metrics.
> >
> > While I agree that fine-tuning is a very effective and widely accepted method, my concern is that fine-tuning LLMs is generally a standard technique and may not offer significant research novelty on its own. It might strengthen your work to further highlight the unique contributions of your approach beyond the application of existing methods to a this domain.

---

> ### Comment · Reviewer_zSTF · 2024-11-24
>
> Regarding A3:
> Thank you for the clarification about the performance metrics.
>
> Regarding A4:
> Thank you for adding the ethics statement.
>
> Regarding A5:
> I am aware of and have seen research showing that LoRA can outperform full model fine-tuning in certain cases. My question is specific to your experiments: why does LoRA demonstrate this behavior in your findings? Some qualitative analysis and discussion is particularly helpful.
>
> I think the explanations are helpful for the community. Thank you!
>
> Regarding A6:
> Have you conducted any experiments using few-shot examples with FlexCAD? It would be helpful to see how FlexCAD performs in a few-shot setting and how it compares to zero-shot settings and vanilla model + few-shots.

---

> > ### Author Response · Authors · 2024-11-25
> > **Thanks for your reply!**
> >
> > We are glad that our responses to Q3 and Q4 have addressed your concerns.
> >
> > **Q8: Some qualitative analysis and discussion between LoRA and Full-model is particularly helpful.**
> > Thanks for your suggestion.
> > As shown in Table 2,
> > LoRA primarily outperforms Full-model in terms of JSD, Unique, and PV metrics,
> > while their performance is nearly identical on the remaining metrics.
> > We hypothesize that this issue is related to **overfitting** and **forgetting pre-trained knowledge**,  as discussed in A5,
> > and include a qualitative performance analysis between LoRA and Full-model in Fig. 19:
> > (1) **Overfitting**. Our statistical analysis indicates that in the training set, the proportion of sketches containing circles or rectangles is as high as 58.4%.
> > As a result, as shown in the blue boxes in Fig. 19, predictions are primarily limited to circles or rectangles (i.e., overfitting to the training set).
> > What's worse, as shown in the red boxes in Fig. 19,
> > Full-model sometimes generates identical or similar circles, which leads to worse Unique and JSD scores (Unique and JSD scores reflect the diversity of the generated samples).
> > Similarly, overfitting may cause the generations for the masked parts to become nearly the same as an element in the training set.
> > Without any changes, these generations may have more possible conflicts with the unmasked part in the test set,
> > such as overlapping between different loops or mismatches between the sketch and existing extrusion operations.
> > Taking sketch-level generation as an example, our statistics show that the proportion of newly generated masked sketches
> > that are the same as a sketch in the training set is 33.7% for Full-model, which is significantly higher than 27.3% for LoRA.
> > As a result, this potentially leads to a lower PV score for Full-model (PV denotes the overall validity of predictions that can be rendered into 3D shapes).
> > Notably, we do not include cases that cannot be rendered into 3D shapes in Fig. 19, as they cannot be rendered into qualitative images either.
> >
> > (2) **Forgetting pre-trained knowledge**. We conduct additional experiments for Full-model, specifically by
> > adding additional instructions (e.g., 'Ensure that the sketch contains at least one circle') at the end of the original instructions.
> > As a result, compared to LoRA, where the proportion of newly generated sketches containing circles increases from 37% to 89% (see A2(5)),
> > Full-model's proportion increases from 37% to 53%.
> > In other words, the instruction-following ability of Full-model is not as strong as that of LoRA.
> > This confirms that, as the number of epochs increases,
> > **pre-trained knowledge may be forgotten during Full-model training**, thus leading to a decline in instruction-following ability.
> > On the other hand, by using LoRA, we can better retain the pre-trained knowledge by freezing most of the parameters.
> >
> >
> >
> >
> >
> > **Q9: Few-shot setting vs. zero-shot settings and vanilla model + few-shots.**
> > A9: Here, we report additional results evaluated on sketch-level controllable generation.
> > The five-shot prompt used is similar to Fig. 18, where all five exemplars are randomly selected from the training set.
> > |Method|COV$\uparrow$|MMD$\downarrow$|JSD$\downarrow$|Novel$\uparrow$|Unique$\uparrow$|PV$\uparrow$|
> > |-|-|-|-|-|-|-|
> > |vanilla (five-shot)|49.5%|1.57|1.60|62.7%|63.5%|57.8%|
> > |FlexCAD (zero-shot,ours)|65.6%|1.19|0.82|__92.1%__|92.6%|93.4%|
> > |FlexCAD (five-shot)|__66.0%__|__1.17__|__0.81__|90.8%|__92.8%__|__94.1%__|
> >
> > The results show that,
> > without fine-tuning, even with few-shot enhancement, the vanilla model performs the worst.
> > Specifically, the output format of the vanilla model is sometimes incompatible with our CAD textual representation.
> > For example, point coordinates for circles, arcs, and lines are sometimes incomplete, and the hierarchy end is missing.
> >
> > Among all models, FlexCAD (five-shot) performs the best in terms of generation diversity and quality.
> > One exception is the Novelty metric (i.e., the percentage of
> > generated CAD models not present in the training set), where FlexCAD (five-shot)
> > slightly underperforms FlexCAD (zero-shot).
> > This could be because the five exemplars used in FlexCAD (five-shot) are drawn from the training set, leading to a higher likelihood of predictions matching those in the training set.
> >
> > We make additional experiments to further study FlexCAD (five-shot).
> > Specifically, we provide five exemplars in which all the answer sketches
> > contain circles. We denote it as FlexCAD (five-shot-all-circle).
> > Compared to FlexCAD (zero-shot) and FlexCAD (five-shot),
> > the proportion of newly generated sketches containing circles for FlexCAD (five-shot-all-circle) increases from 35%-37% to 65%.
> > This further demonstrates that our model can generate CAD models guided by few-shot learning.

---

> > ### Author Response · Authors · 2024-11-30
> > **Status Update Request for Paper Review**
> >
> > Dear Reviewer zSTF,
> >
> > We hope that the additional results and discussions provided address your concerns. As the deadline for Author-Reviewer discussion approaches, we eagerly look forward to your responses.  Please feel free to let us know if there are any further clarifications we can offer. We would be happy to continue the discussion if any issues remain.
> >
> > Thank you for your time and consideration.
> >
> > Best,
> > Authors

---

> ### Author Response · Authors · 2024-11-25
> **Thanks for your reply!**
>
> **Q7: Highlight the unique contributions of your approach beyond the application of existing methods.**
> A7: (1) Representing CAD as concise and structured text is a unique contribution (see Section 3.1).
> In standard LLM fine-tuning, the text format that can be
> easily processed by LLM is naturally existing, such as natural language or
> Python code.
> However, existing CAD representation cannot be easily understood by LLMs.
> In prior works, Hnc-cad and SkexGen both use one-hot vectors to represent tokens in CAD sequences.
> Specifically, SkexGen employs a one-hot vector with a dimension of 4101 to represent a coordinate token, while Hnc-cad adopts the same representation.
> To address this limitation, we propose representing CAD models as concise and structured texts, making them more accessible for LLMs.
> (2) Our hierarchy-aware masking mechanism is another unique contribution (see Section 3.2).
> In standard LLM fine-tuning, randomly masking a continuous set of tokens is generally utilized.
> In contrast, we mask a specific hierarchy for each CAD text at each epoch.
> Additionally, we incorporate hierarchy-level information into the mask token (e.g., [face mask]).
> As shown in Table 3, these designs help LLMs focus more effectively on generating local hierarchies, thereby improving both generation quality and diversity.
>
> Benefiting from (1) and (2), as mentioned in Lines 60-65, our LLM-based FlexCAD surpasses SkexGen and Hnc-cad, which have limited controllability,
> by enabling controllable CAD generation across all hierarchies, including CAD, sketch-extrusion, extrusion, sketch, face, loop, and curve.

---

### Author Response · Authors · 2024-11-19
**To all Reviewers**

Dear Reviewers,

We sincerely thank you for your valuable comments.
We are encouraged that you found our method **novel** [4Ahh], **bridging LLMs and CAD models** [4Ahh],
offering **a promising insight for future CAD design** [DaWS].
We are also pleased that you found our approach to be effective [zSTF, DaWS, 4Ahh],
improving generation quality and controllability [zSTF], well-organized [DaWS], clearly written [DaWS], and easy to follow [DaWS].

In the following, we will do our best to address the misunderstandings.
Additionally, our manuscript has been revised to add Fig. 18 (a five-shot in-context learning prompt example for GPT-4o) and the Ethics Statement section,
based on the reviewers’ insightful comments, making our research more robust and accessible.

As the deadline for Author-Reviewer discussion approaches,
we eagerly look forward to your responses.
Please let us know if there are any additional clarifications or experiments we could provide.
We would be happy to discuss further if any concerns remain.
Thanks again for your time!

Best,

Authors

---

### Meta-Review · Area_Chair_2wqo · 2024-12-20

**Metareview:**

This paper proposes FlexCAD, a unified model for controllable CAD generation across construction hierarchies using fine-tuned LLMs. By representing CAD models as structured text and employing a hierarchy-aware masking strategy, it bridges CAD's complex numerical data with LLMs' textual capabilities, enabling unified generation without separate models.

A notable strength of this paper is its pioneering application of LLMs to CAD generation. While fine-tuning LLMs is well-established, applying it to CAD—characterized by complex geometric data and hierarchies—demonstrates that LLMs can effectively handle structured CAD text, enabling new interactions and control beyond traditional methods. The authors demonstrate promising improvements in generation quality and controllability as an early attempt to apply LLMs to CAD generation.

While the technical novelty in model architecture may be limited, applying LLMs to CAD generation marks a significant advancement in a specialized field. Representing CAD models as structured text for LLMs is a meaningful contribution, making complex CAD data accessible to language models despite limited improvements in spatial understanding and data representation. This paper adapts LLMs for CAD generation, opening new avenues in design automation and encouraging further exploration.

**Additional Comments On Reviewer Discussion:**

During the review process, Reviewer zSTF questioned the paper's technical novelty, doubting whether fine-tuning LLMs for CAD without substantial advancements was sufficient. In response, the authors emphasized that their structured text representation and hierarchy-aware masking strategy enable LLMs to effectively handle complex CAD data and support controllable generation across all hierarchies, representing a meaningful contribution to the field.

Reviewers 4Ahh and DaWS raised key points, questioning the advantages of LLMs over traditional methods and suggesting evaluation on additional datasets. The authors demonstrated FlexCAD's intuitive interactions through inherited abilities like instruction following and noted the limited availability of large-scale CAD datasets. Both reviewers were positive and acknowledged the potential benefits and innovative aspects of this approach.

Despite some concerns, the paper's approach to data representation and controllable CAD generation offers valuable contributions, supporting its acceptance for advancing LLM applications in specialized domains like CAD.

---

### Decision · Program_Chairs · 2025-01-22

Accept (Poster)